# Histones and histone variant families in prokaryotes

Samuel Schwab [1,2,3], Yimin Hu [4], Bert van Erp[1,2,3], Marc K. M. Cajili[1,2,3], Marcus D. Hartmann [4,5], Birte Hernandez Alvarez [4], Vikram Alva [4], Aimee L. Boyle[1,2,3,6] & Remus T. Dame [1,2,3] ✉

Histones are important chromatin-organizing proteins in eukaryotes and archaea. They form superhelical structures around which DNA is wrapped. Recent studies have shown that some archaea and bacteria contain alternative histones that exhibit different DNA binding properties, in addition to highly divergent sequences. However, the vast majority of these histones are identified in metagenomes and thus are difficult to study in vivo. The recent revolutionary breakthroughs in computational protein structure prediction by AlphaFold2 and RoseTTAfold allow for unprecedented insights into the potential function and structure of previously uncharacterized proteins. Here, we categorize the prokaryotic histone space into 17 distinct groups based on AlphaFold2 predictions. We identify a superfamily of histones, termed α3 histones, which are common in archaea and present in several bacteria. Importantly, we establish the existence of a large family of histones throughout archaea and in some bacteriophages that, instead of wrapping DNA, bridge DNA, thereby diverging from conventional nucleosomal histones.

Cells from the two domains of life, the archaeal and bacterial domains, use chromatin proteins to structure and organize their genomic DNA. This stems from the basic necessity of fitting the chromosomal DNA into the limited cell volume. Without any compaction, the chromosomal DNA takes up 1000x the cell volume, illustrating the necessity of chromatin proteins in genome organization. Chromatin proteins are diverse with no single group being common in all domains of life. However, their architectural properties are largely conserved. Chromatin proteins are roughly divided into three groups based on how they structure DNA: proteins that locally deform DNA by bending or wrapping it, those that bridge DNA across short and long distances, and those that form long filaments on DNA[1]. In eukaryotes, structural maintenance of chromosomes (SMC) proteins facilitate long-range interactions by bridging DNA while histones bend and wrap DNA. The four core histones, H2A, H2B, H3, and H4, are universal to all eukaryotes and together form the octameric nucleosome which wraps 150

base pairs of DNA[2]. While these core histones diverge largely in sequence, they share the same fundamental histone fold. This histone fold consists of approximately 65 amino acids that form three alpha helices linked together by two short linkers. Positively charged tails are present at the N-terminus and are targets for post-translational modifications. The smallest functional units of histones are dimers, which bind and bend 30 base pairs of DNA. In eukaryotes, H2A, H2B, H3, and H4 form H2A/H2B and H3/H4 heterodimers. The formation of heterodimers instead of homodimers is thought to be promoted by the N-terminal tails[3]. The nucleosome is formed by consecutively linking H2A/H2B and H3/H4 heterodimers into a superhelix.

Histones are not exclusive to eukaryotes. They are also present in the majority of archaeal phyla. The model archaeal histones are HMfA and HMfB from *Methanothermus fervidus* and HTkA and HTkB from *Thermococcus kodakarensis*[4–11]. Archaeal histones differ from their eukaryotic counterparts in that they generally lack tails and can form

[1]Leiden Institute of Chemistry, Leiden University, Leiden, The Netherlands. [2]Centre for Microbial Cell Biology, Leiden University, Leiden, The Netherlands. [3]Centre for Interdisciplinary Genome Research, Leiden University, Leiden, The Netherlands. [4]Department of Protein Evolution, Max Planck Institute for Biology Tübingen, Tübingen, Germany. [5]Interfaculty Institute of Biochemistry, University of Tübingen, Tübingen, Germany. [6]School of Chemistry, University of Bristol, Bristol, UK. ✉e-mail: rtdame@chem.leidenuniv.nl

homodimers. However, histones of the Asgard archaea, a sister clade of eukaryotes, often contain eukaryotic-like positively charged N-terminal tails. Whether these archaeal tails are the ancestors of eukaryotic histone tails is unknown[12]. Archaeal histones form similar nucleosomal structures called hypernucleosomes[13–16]. While eukaryotic nucleosomes are limited to an octamer, archaeal hypernucleosomes can theoretically extend infinitely. How the size or positioning of hypernucleosomes is regulated is poorly understood. HMfB shows a preference for sequences with alternating A/T and G/C-rich regions, similar to the eukaryotic histones[17]. An artificial high-affinity sequence, "Clone20", which promotes the formation of tetrameric nucleosomes, was identified for HMfA and HMfB. The biological relevance of the Clone20 sequence remains uncertain as similar sequences have not yet been found in archaeal genomes[17,18]. Histones have, until recently, been considered to only wrap DNA. However, histone MJ1647 from *Methanocaldococcus jannaschii* diverges from conventional histones in that it bridges DNA[19]. MJ1647's bridging ability facilitates long-range DNA interactions, a feature not found in any other histone. These findings suggest that more histones might exist with alternative DNA-binding properties.

Bacteria and viruses are generally considered to lack histones. Instead, bacteria contain nucleoid-associated proteins (NAPs). Although NAPs are structurally different from classical histones, they are considered histone-like because they are generally small, highly expressed proteins with roles functionally similar to histones. However, the protein-DNA complexes that NAPs form differ. While histones strongly wrap DNA, NAPs generally bend, bridge DNA, or form protein-DNA filaments. Prime examples are histone-like nucleoid structuring protein (H-NS) from *Escherichia coli* which bridges DNA and forms protein-DNA filaments, integration host factor (IHF) which bends DNA at an angle of almost 180°, and factor for inversion stimulation (Fis) which bends DNA at a shallower angle of 60°[20].

Histones are not altogether absent in bacteria and viruses[21,22]. Recent studies have shown that some do indeed use histones to organize their DNA. The viral family *Marseilleviridae* contains double histone variants of the core eukaryotic histones[23]. These viral double histones contain two histone folds fused into one protein chain. They form nucleosomes which are, structurally, almost identical to eukaryotic nucleosomes. Furthermore, these viral nucleosomes are used to package viral DNA within the virions[24]. In bacteria, *Bdellovibrio bacteriovorus* expresses an essential histone, HBb (locus name: Bd0055)[25,26]. Compared to eukaryotic and archaeal histones, it forms vastly different quaternary structures. The first co-crystal structure suggested that HBb forms long protein-DNA filaments and does not bend or wrap DNA[25]. However, in a subsequent study, a different co-crystal structure, in-solution data, and molecular dynamics data show that HBb bends DNA as a dimer, similar to members of the HU/IHF protein family[26].

The histones from *B. bacteriovorus* and *M. jannaschii* are the first examples of histones with alternative architectural properties, from the conventional wrapping of (hyper)nucleosomal histones to bridging and bending. Interestingly, although these histones have a completely different DNA binding mode, their histone folds are almost identical to those of nucleosomal histones. This suggests that the histone fold is a fundamental chromatin protein fold that can be adapted to different architectural needs with minor modifications. The enormous growth of available metagenomic sequencing data over the last decade provided an unprecedented view of protein diversity throughout life. However, until recently the lack of structural data had limited the functional analysis of protein homologs. With the release of AlphaFold2 and RoseTTAFold, we are now able to reliably predict monomeric and multimeric structures with atomic accuracy[27–29]. We set out to predict monomer and multimer structures of all prokaryotic histone fold proteins in the UniProt database. Here, we map out a largely undiscovered prokaryotic histone world, categorizing all prokaryotic histones into 17 distinct groups. Alongside nucleosomal histones, we identify a second major prokaryotic histone family, termed α3 histones, with representatives in almost all archaeal phyla and several bacteria. Furthermore, we discovered various histones that bridge DNA, demonstrating that histones with alternative architectural properties are widespread in prokaryotes.

## Results

### Identifying prokaryotic histones

To find histones in archaea and bacteria, we used the protein annotation database InterPro[30,31]. InterPro classifies all proteins within the UniProt database into families, domains, and important sites based on their sequences. For histones, InterPro contains the histone-fold homologous superfamily, which serves as a comprehensive category for proteins exhibiting the histone fold. We retrieved all sequences that are part of this superfamily. To validate that these sequences indeed contain a histone fold and to gain insight into their potential function, we predicted monomer and multimer structures with AlphaFold2[27,28]. The majority of our predictions have high confidence scores and deep multiple-sequence alignments (Supplementary Figs. 1–4). A general explanation on using AlphaFold2 and interpreting the confidence values is provided in the Supplementary. We identified a total of 5823 histones in prokaryotes, 25% of which are from bacteria. Half of the 5823 histones have not been previously identified. We refer to all histones as "histones" as they are all predicted to feature the characteristic histone fold structure (Fig. 1a). However, it is not clear whether they all function as global genome organizers. Histones form dimers in solution, whereby the α2 helices cross each other and the α1 and α3 helices are positioned on opposite faces of the dimer (Fig. 1b). The dimer represents the smallest functional unit and can bind 30 base pairs of DNA at its α1 face. Eukaryotic histones form nucleosome structures by linking dimers consecutively through their α3 helices and

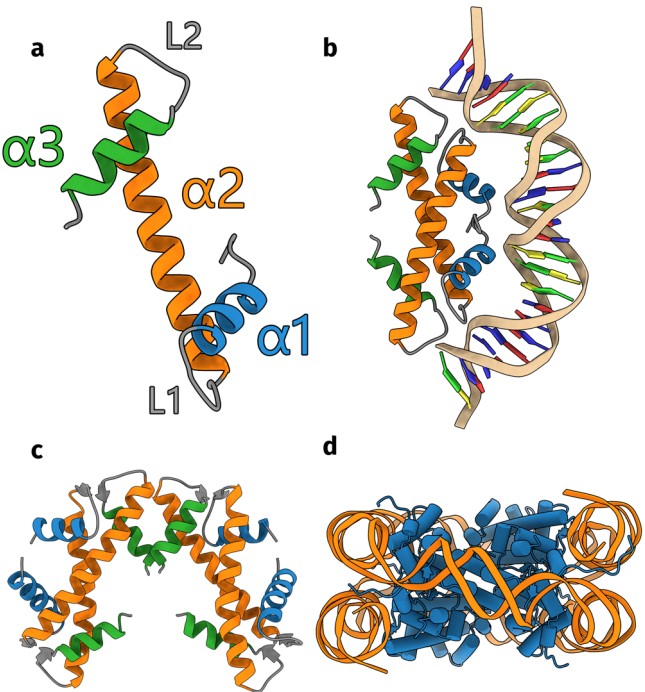

**Fig. 1 | The conventional histone protein forms nucleosomes. a** The histone fold (PDB: 1A7W[9]). The N-terminal α1, central α2, and C-terminal α3 helices are colored blue, orange, and green respectively. Linkers L1 and L2 connect the α1 and α3 helices to the central α2 helix. **b** The histone dimer binds DNA at its α1 face (PDB: 5T5K[13]). **c** The eukaryotic H3-H4 tetramer (PDB: 1AOI[2]). Only the core histone fold is visualized. **d** The eukaryotic octameric nucleosome (PDB: 1AOI[2]).

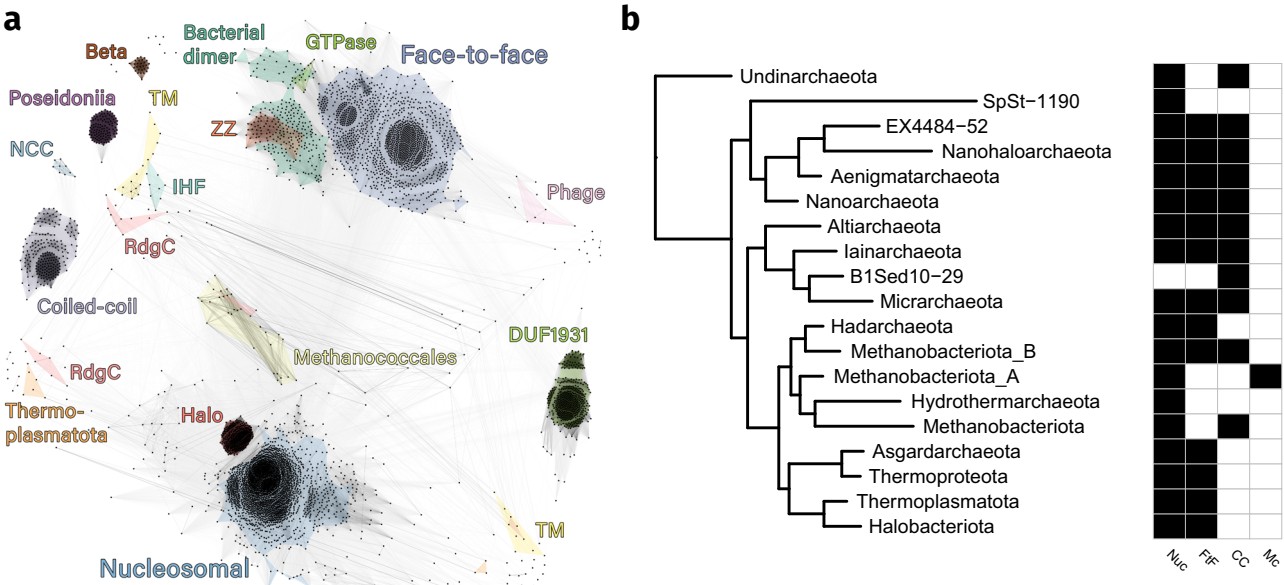

**Fig. 2 | Alternative histones are diverse and found across prokaryotes.**
**a** Clustered sequence space of prokaryotic histones. Clustering was performed with CLANS. The color of each line indicates the sequence similarity between the two sequences; sequences that are connected by darker lines are more similar than those connected by lighter lines. Clusters are colored based on the histone category to which they belong as determined by the AlphaFold2 predictions. For a short description of each histone category, see Supplementary Table 2. **b** Clado-gram of archaea showing the distribution of nucleosomal (Nuc), face-to-face (FtF), coiled-coil (CC), and Methanococcales (Mc) histones across different phyla. The cladogram is based on GTDB version 207. For reference, Methanobacteriota A contains the Methanopyri and Methanococci classes; Methanobacteriota B con-tains the Thermococci and the Methanofastidiosa classes.

the last 8 C-terminal residues of their α2 helices, and possess long N-terminal tails (Fig. 1c, d). These tails are generally lacking in archaeal nucleosomal histones, although we find several nucleosomal histones in the Asgard archaea phylum that have long disordered N-terminal tails (Supplementary Fig. 5). Most new histones differ significantly from conventional nucleosomal histones in sequence and quaternary structure. We manually reviewed every prediction and its confidence values. We grouped histones together if they are predicted to form similar quaternary structures and if the predicted interface for this multimer has low predicted aligned errors (<10Å) or if the histones share similar additional domains. Histones that are similar in sequence to the recently described bacterial histone Bd0055/HBb were classified into the group "bacterial dimer histones" as they lack unique qua-ternary structures. In this manner, we subdivided all prokaryotic his-tones into 17 categories, 13 of which have not been identified before (Supplementary Table 2). To visualize the sequence space of prokar-yotic histones, we clustered all histone sequences based on their all-against-all pairwise sequence similarity in a two-dimensional space using the CLANS software[32] (Fig. 2a). The histone categories deter-mined by structure prediction are visualized in the CLANS map; they overlap well with the sequence similarity-based clusters.

Based on sequence and structure similarity, we have identified two major prokaryotic histone families: the nucleosomal histones and the α3-truncated histones, referred to from now on as α3 histones. The α3 family compromises five histone categories, each of which is likely related to a different function as they differ in predicted quaternary structure and the presence of additional domains: face-to-face, bac-terial dimer, ZZ, Rab GTPase, and phage histones. These categories are discussed in more detail below. α3 histones are defined by a truncated α3 helix, which, compared to the 10 amino acid long nucleosomal α3 helix, consists only of 3 to 4 amino acids. Furthermore, their α2 helix is 4 to 5 amino acids shorter compared with nucleosomal histones. α3 histones are related in structure and sequence, as the five categories that make up the superfamily cluster close together in the CLANS map (Fig. 2a). While nucleosomal histones exist exclusively in archaea,

about 40% of α3 histones are from bacteria (Supplementary Fig. 6). However, they are not well conserved within the bacterial domain as only 1.15% of bacterial proteomes in UniProt contain an α3 histone, in contrast to archaea where α3 histones are found in almost all phyla. While nucleosomal and α3 histones present the two major families, only 65% of histones belong to either of these two families. The remaining histones are part of minor, highly diverse histone cate-gories. Some minor histones appear to have dramatically changed their architectural properties, possibly bridging DNA instead of wrapping it, based on their predicted multimer structures. Other minor histones appear to have lost their DNA binding ability, as they lack identifiable DNA-binding residues and have instead gained trans-membrane domains (Supplementary Fig. 7). In the subsequent sec-tions, we will go further into detail about some of these histones, focussing on the most prominent histone families, the α3 histones, and histones that likely bridge DNA.

**Face-to-face histones**
Face-to-face (FtF) histones make up the largest subcategory of α3 histones (83%) and are, after nucleosomal histones, the largest group of histones in prokaryotes. FtF histones are found in the majority of archaeal phyla (Fig. 2b). In bacteria, FtF histones are predominantly found in the phyla Spirochaetota, Planctomycetota, Bdellovibrionota (class Bacteriovoracia), and Myxococcota (Supplementary Figs. 6 and 8). FtF histones are defined by their predicted tetramer structure (Fig. 3a). In the tetramer, two dimers interact in a manner similar to nucleosomal histones, via their α3 helices and the last -8 amino acids of their α2 helix. Unlike nucleosomal histones, both sides of the dimer interact, forming a torus. To gain more insight into important residues, we aligned all FtF histones, constructed an HMM profile, and visualized this profile as an HMM logo (Supplementary Fig. 9). The HMM logo shows which residues are strongly conserved between FtF histones and are probably of functional importance. The strongest conserved residues are found at the C-terminus of the pro-tein and include residues 48, 52, 54, 56, and 61 in the HMM profile

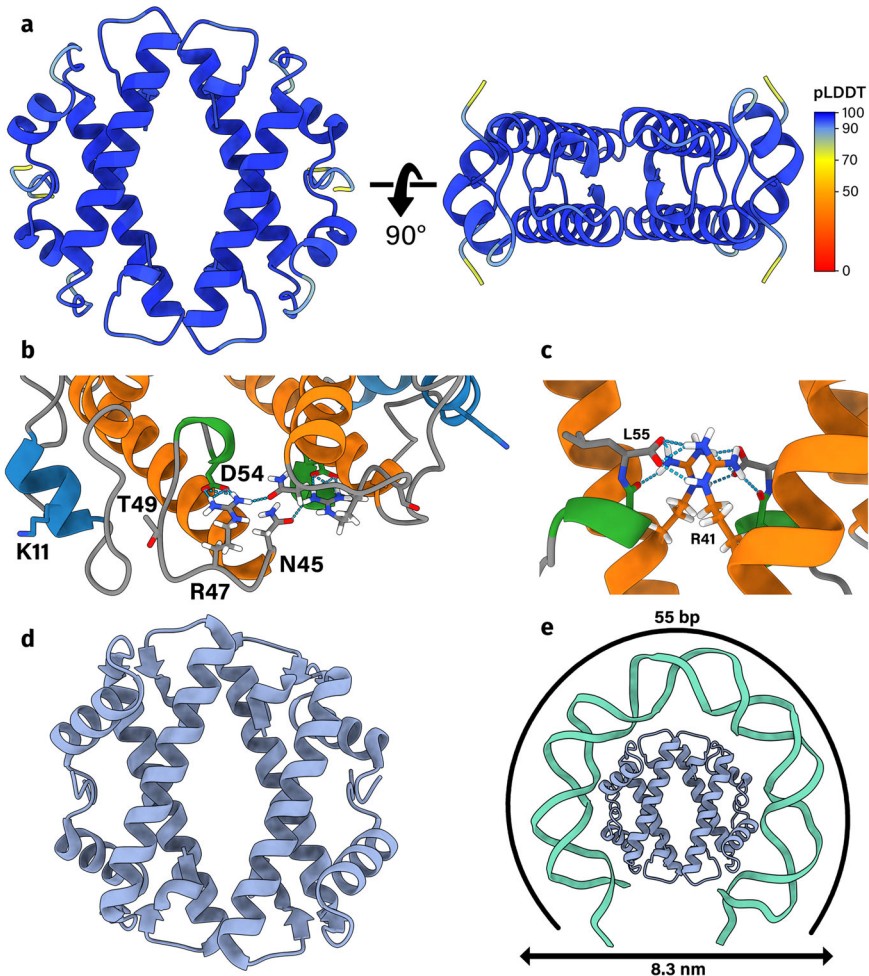

**Fig. 3 | The face-to-face (FtF) histones form a unique tetramer structure. a** The homotetramer of FtF histone D4GZE0 from *Haloferax volcanii* as predicted by AlphaFold2. Each residue is colored by its predicted local distance difference test (pLDDT) value. AlphaFold2 is confident in the local structure if the pLDDT is >70. **b** The conserved RxTxxxxD motif, DNA binding residues K11, and tetramerization residue N45 of FtF histone D4GZE0. K11, N45, R47, T49, and D54 relate to K16, N52, R54, T56, and D61 in the FtF HMM logo (Supplementary Fig. 9). **c** The `back' of the dimer-dimer interface of FtF histone D4GZE0. R41 relates to R48 in the FtF HMM logo (Supplementary Fig. 9). **d** Crystal structure of FtF histone HTkC from *Thermococcus kodakarensis* (PDB: 9F2C). **e** Our proposed model for how FtF histones bind and wrap DNA.

(Supplementary Fig. 9). These conserved residues correspond to R41, N45, R47, T49, and D54 for FtF histone D4GZE0 from the archaeal model organism *Haloferax volcanii* (Fig. 3b, c). Residues R47, T49, and D54 form an RxTxxxxD motif which is also present in nucleosomal histones from archaea (Supplementary Fig. 10). The arginine and the aspartate are responsible for structuring the L2 loop, while the threonine, located in the L2 loop, is involved in DNA binding. Compared to nucleosomal histones, the FtF histone HMM profile lacks a strongly conserved DNA-binding arginine or lysine at position 55 (RKTxxxxD) (Supplementary Fig. 9). Residues R41 and N45 are responsible for the dimer-dimer interactions at the dyad. In the predicted D4GZE0 structure, N45 is located at the 'front' of the dimer-dimer interface and can form hydrogen bonds with R47 of the opposing dimer. R41 is located further back within the dimer-dimer interface and forms salt bridges with the carboxyl group of residue 55 of the opposing dimer (Fig. 3c). On the α1 helix, a lysine is strongly conserved at position 16 (Supplementary Fig. 9). This residue corresponds to K11 for FtF histone D4GZE0 (Fig. 3b). Conserved lysines can also be found at positions 12, 14, and 20 in the HMM profile, although with lower frequency. As these lysines are solvent-exposed, conserved, and positively charged, they are likely involved in DNA binding.

To confirm the predicted tetramer structure of FtF-histones, we purified and crystallized the FtF histone from *Thermococcus*

*kodakarensis*, which we refer to as HTkC. We solved the crystal structure at a resolution of 1.84 Å (PDB: 9F2C) via molecular replacement using the predicted structure as a search model (Fig. 3d). The asymmetric unit contains two histone dimers, that assemble into a torus-shaped tetramer. With a Cα RMSD of 0.652 Å, the HTkC crystal structure is virtually identical to the AlphaFold prediction (Supplementary Fig. 11), highlighting AlphaFold's accuracy in predicting histone structures.

To gain more insight into the possible function of FtF histones, we examined transcriptome data from Halobacteria, Thermococci, and Leptospirales, three well-studied taxa where FtF histones are common. Halobacteria contain one FtF histone on their chromosome and additional ones on their plasmids. The chromosomal FtF histones in *H. volcanii* and *Halobacterium salinarum*, HVO0196 and VNG2273H, respectively, are among the top 2% of the highest expressed genes across three out of four transcriptome datasets (Supplementary Fig. 12a–d). As the FtF histone is highly expressed, it is a likely candidate to be the unknown protein that causes the nucleosome-like organization of chromatin in Halobacteria. Electron microscopy images of the chromosomal fibers of *H. salinarum* show beads-on-a-string-like structures[33]. These beads were estimated to have a diameter of $8.1 \pm 0.6$ nm, somewhat smaller than the eukaryotic nucleosomes of 11 nm. Micrococcal nuclease digestion of crosslinked *H. volcanii*

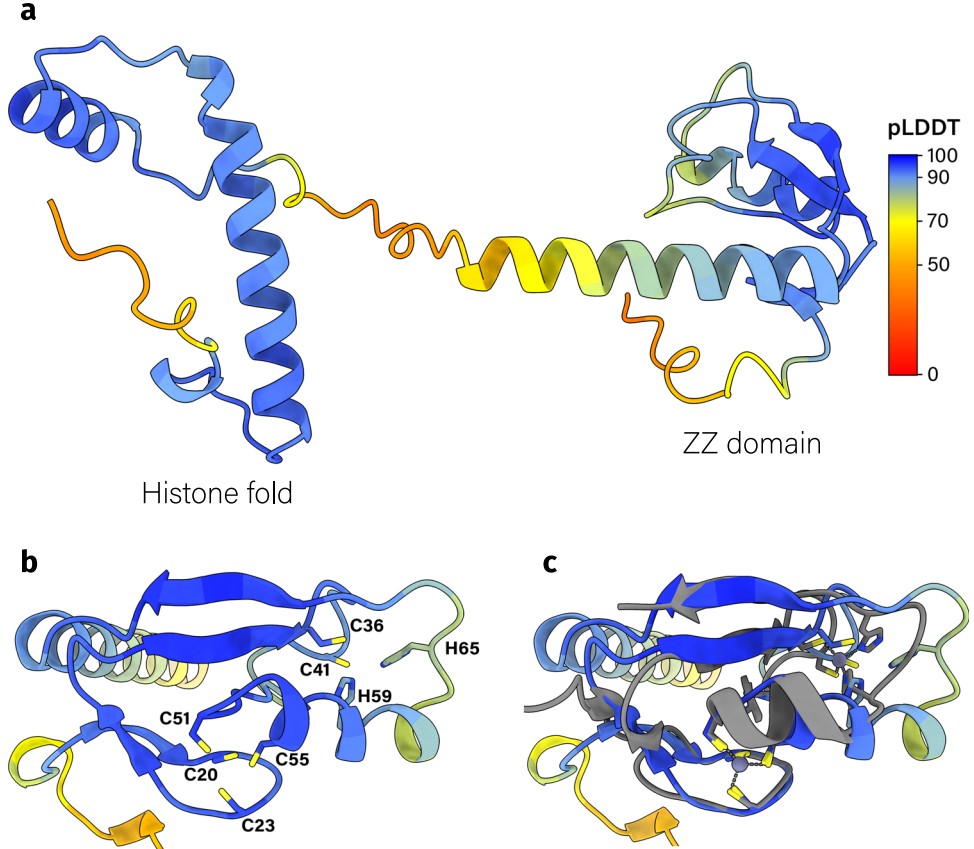

**Fig. 4 | ZZ-type zinc finger histones can possibly bind two zinc ions. a** The ZZ-histone D0LYE7 from *Haliangium ochraceum SMP-2* as predicted by AlphaFold2. ZZ-histones contain a ZZ-type zinc finger domain at the N-terminus and an *α*3 histone fold at the C-terminus. Each residue is colored by its pLDDT value. **b** The ZZ domain of D0LYE7. This domain contains two zinc binding motifs, one C4 motif (C20, C23, C51, C53) and a C2H2 motif (C36, C41, H59, H65). These residues correspond to C10, C13, C41, and C43 for the C4 motif and C26, C31, H49, and H55 for the C2H2 motif in the HMM profile (Supplementary Fig. 18). Each residue is colored by its pLDDT value. **c** The ZZ domain of HERC2 (gray, PDB: 6WW4[35]) aligned to D0LYE7.

chromatin showed protected DNA fragments of 50 to 60 base pairs[34], suggesting that this unknown protein binds 50 to 60 base pairs of DNA. The expression of this unknown protein is expected to be high as the *H. volcanii* genome is estimated to contain 14.2 nucleosomes per kilobase, 2.7 times higher than the 5.2 nucleosomes per kilobase in *Saccharomyces cerevisiae*[34]. Not all organisms with FtF histones show such high expression levels. The FtF histone, HTkC, in *T. kodakarensis* is part of the top 7% of the highest expressed genes, with an expression level that is 34 times lower than that of hypernucleosome histone HTkA (Supplementary Fig. 12e). The transcriptome of the related organism *Thermococcus onnurineus* was measured in three different conditions: in yeast extract-peptone-sulfur (YPS), modified minimal-CO (MMC), and modified minimal-formate (MMF) media (Supplementary Fig. 12f–h). In YPS and MMF, the FtF histone shows a two to three times lower expression level than the other two nucleosomal-like histones, B6YSY3 and B6YXB0. In MMF, however, the FtF histone is expressed twice as highly as the nucleosomal histones, indicating that environmental factors might play a role in regulating the expression of FtF histones in hypernucleosome-containing archaea. Similar to Halobacteria, the FtF histone in the pathogenic bacterium *Leptospira interrogans serovar Lai* is among the top 2% of the highest expressed genes (Supplementary Fig. 12i). As we find no other known NAPs among the top 10% of expressed genes, the FtF histone might be the main architectural protein in Leptospirales. The FtF histone is essential to *Leptospira interrogans serovar Lai*, as attempts to delete the gene from its genome have failed[25]. The histone is found in both pathogenic and free-living saprophytic Leptospirales and always contains an N-terminal tail. Tails are common in bacterial FtF histones as opposed to archaeal FtF histones, which generally lack tails (Supplementary Fig. 13). There is no conserved tail sequence across bacteria; however, the majority of them are positively charged.

Based on the aforementioned studies on *H. volcanii*, we propose a model for the binding of FtF histones to DNA. If the FtF histone is indeed the main architectural protein of Halobacteria, it binds 50 to 60 base pairs of DNA and forms beads-on-a-string-like structures with diameters of 8.1 ± 0.6 nm. To fit 50 base pairs, the DNA would have to wrap around the tetramer, similarly to nucleosomal histones (Fig. 3e). Furthermore, the diameter of this DNA-wrapped tetramer would be 8.3 nm, similar to the observed beads-on-a-string-like structures. However, one possible issue with this model is steric hindrance caused by the bending of DNA molecules close to each other.

### Minor *α*3 histones

In addition to FtF histones, within the *α*3 histone family, there are four, smaller, categories: bacterial dimers, ZZ histones, Rab GTPase histones, and phage histones. The bacterial dimers are predominantly found in Bdellovibrionota (class Bdellovibrionia), Elusimicrobiota, Spirochaetota (class Spirochaetia), Planctomycetota, Myxococcota_A, and Chlamydiota (Supplementary Figs. 6b and 8). AlphaFold does not produce a confident multimer prediction larger than dimers for these histones, hence the name 'bacterial dimers' (Supplementary Fig. 14a). Closer inspection of the HMM logo shows that bacterial dimers lack the conserved residues R48 and N52, which facilitate the dimer-dimer interaction in FtF histones (Supplementary Fig. 15). Similar to

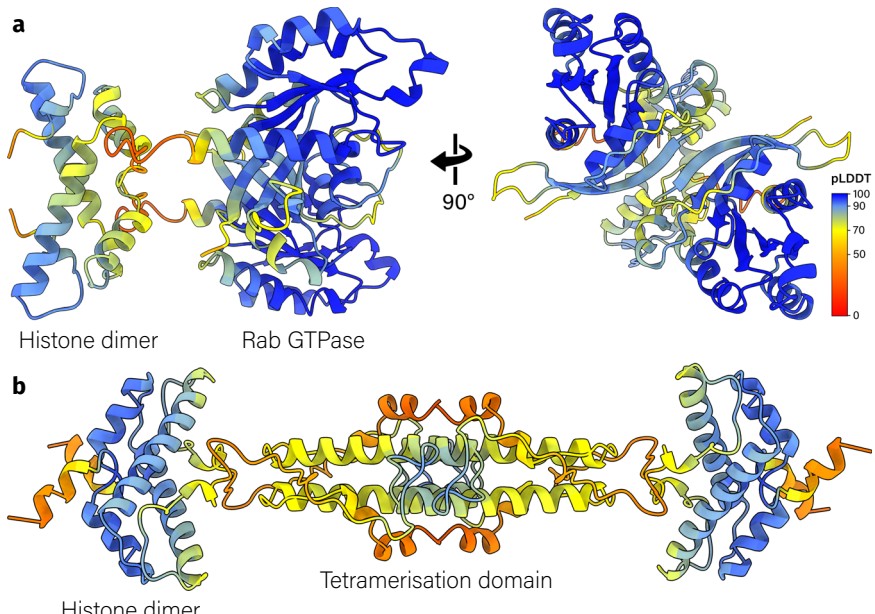

**Fig. 5 | α3 histones from bacteriophages or with eukaryotic-like domains. a** The homodimer of Rab GTPase histone A0A0F8XJF6 as predicted by AlphaFold2. Rab GTPase histones contain a small Rab GTPase domain on the N-terminus and an FtF-like histone fold at the C-terminus. Each residue is colored by its pLDDT value.

**b** The homotetramer of phage histone A0A2E7QIQ9 as predicted by AlphaFold2. Phage histones contain an α3 histone fold at the N-terminus and an α-helix which functions as the tetramerization domain at the C-terminus.

nucleosomal histones, we find the RKTxxxD motif in bacterial dimers (Supplementary Fig. 14b). They also contain conserved lysines on their α1 helix at positions 11, 13, and 17, which possibly bind DNA. K17 is also conserved in nucleosomal histones as is an arginine (R20), while K13 is conserved in FtF histones (K16) (Supplementary Figs. 9, 10, and 15). The bacterial dimer HBb (locus name: Bd0055) in *Bdellovibrio bacteriovorus* HD100 is highly expressed, being part of the top 6% of highest expressed genes during its growth phase with an absolute expression level similar to IHF, HU, and SMC (Supplementary Fig. 16a). Furthermore, proteomics data show that HBb is highly abundant in the attack phase[25]. Based on crystal structures, in vitro characterization, and molecular dynamics data, we have recently demonstrated that HBb bends DNA, similar to members of the HU/IHF protein family[26].

ZZ histones, which are closely related to bacterial dimers, are predominantly found in Proteobacteria (Supplementary Fig. 8). They consist of a ZZ-type zinc finger domain at their N-terminus and a bacterial-dimer-like histone domain at their C-terminus (Fig. 4a). The ZZ-domain contains two conserved zinc-binding sites, a C4 and a C2H2 site, and is structurally similar to the eukaryotic ZZ-domains of HERC2 (a ubiquitin protein ligase), p300 (a histone acetyltransferase), and ZZZ3 (a histone H3 reader) (Fig. 4b–c and Supplementary Figs. 17–18). These eukaryotic proteins all participate in the post-translational modifications of histones. Intriguingly, these eukaryotic ZZ-domains bind to the tail of histone H3, potentially aiding their localization to nucleosomes[35–37]. If the function of the ZZ-domain is conserved, it may bind H3-like tails in bacteria too. In fact, H3-like positively charged tails are found in the majority of bacterial FtF histones (Supplementary Figs. 13, 19 and 20). However, the binding pocket residues of the eukaryotic ZZ-domain are not conserved in the bacterial variant (Supplementary Fig. 21). Given that only 50% of the proteomes that have a ZZ histone also contain a second histone and that bacterial histone tails lack a conserved sequence, it seems unlikely that the ZZ-domain binds to tails of other histones. Transcriptome data from *B. bacteriovorus* HD100 reveal very low expression of the ZZ histone during both attack and growth phases (top 43 to 70% of the highest expressed genes) (Supplementary Fig. 16). Similarly, proteomics data indicate a low abundance of the ZZ histone during the attack phase as well as in the host-independent strain HID13, which lacks the attack phase[25].

The last two minor α3 histone types are the Rab GTPase and phage histones. Rab GTPase histones are found exclusively in Lokiarchaeota. They contain a FtF-like histone at their C-terminus and a Rab GTPase domain, a subfamily of Ras GTPases involved in regulating membrane trafficking pathways, at their N-terminus (Fig. 5a). The closest homologs of these Rab GTPase domains are the small Rab GTPases from eukaryotes. GTPase-related histones, which are also found in eukaryotes, include the eukaryotic Ras activator Son of Sevenless (SOS) that contains a double histone domain[38]. This histone domain binds to lipids and regulates SOS activity to activate Ras GTPases[39,40]. Whether the archaeal Rab GTPase histones similarly bind to lipids is unclear as the histone domain in SOS has no detectable sequence similarity with the Rab GTPase histones. Structurally the SOS histone domain is more similar to the H2A/H2B heterodimer than to the Rab GTPase histones. The last minor α3 histone type, the phage histones, are found in prokaryotic dsDNA virus metagenomes and bacterial metagenomes. In the viral metagenomes we find tail proteins, suggesting that these bacteriophages are part of the Caudovirales order. Since some bacterial metagenomes also contain the histone, the bacteriophage might be a prophage. The phage histones contain an α3 histone fold at their N-terminus and an alpha-helical domain at their C-terminus. In the tetramer prediction, the C-terminal domains form a tetramerisation domain (Fig. 5b). The histone domain lacks the RxTxxxD motif and shows low sequence identity with other histone types (Supplementary Fig. 22). It contains two conserved residues that possibly bind DNA: K10, and K49, which correspond to the DNA binding residues K20 and K63 in the nucleosomal HMM logo (Supplementary Fig. 10). Interestingly, one of the highest conserved residues is an arginine on the side of the histone dimer (Supplementary Fig. 23). As the phage histone is predicted to form a tetramer structure through its C-terminal domain instead of through the histone folds, it may bridge two DNA duplexes.

## DNA bridging histones
Among the 17 prokaryotic histone categories, the predicted quaternary structures of four suggest potential DNA bridging capabilities.

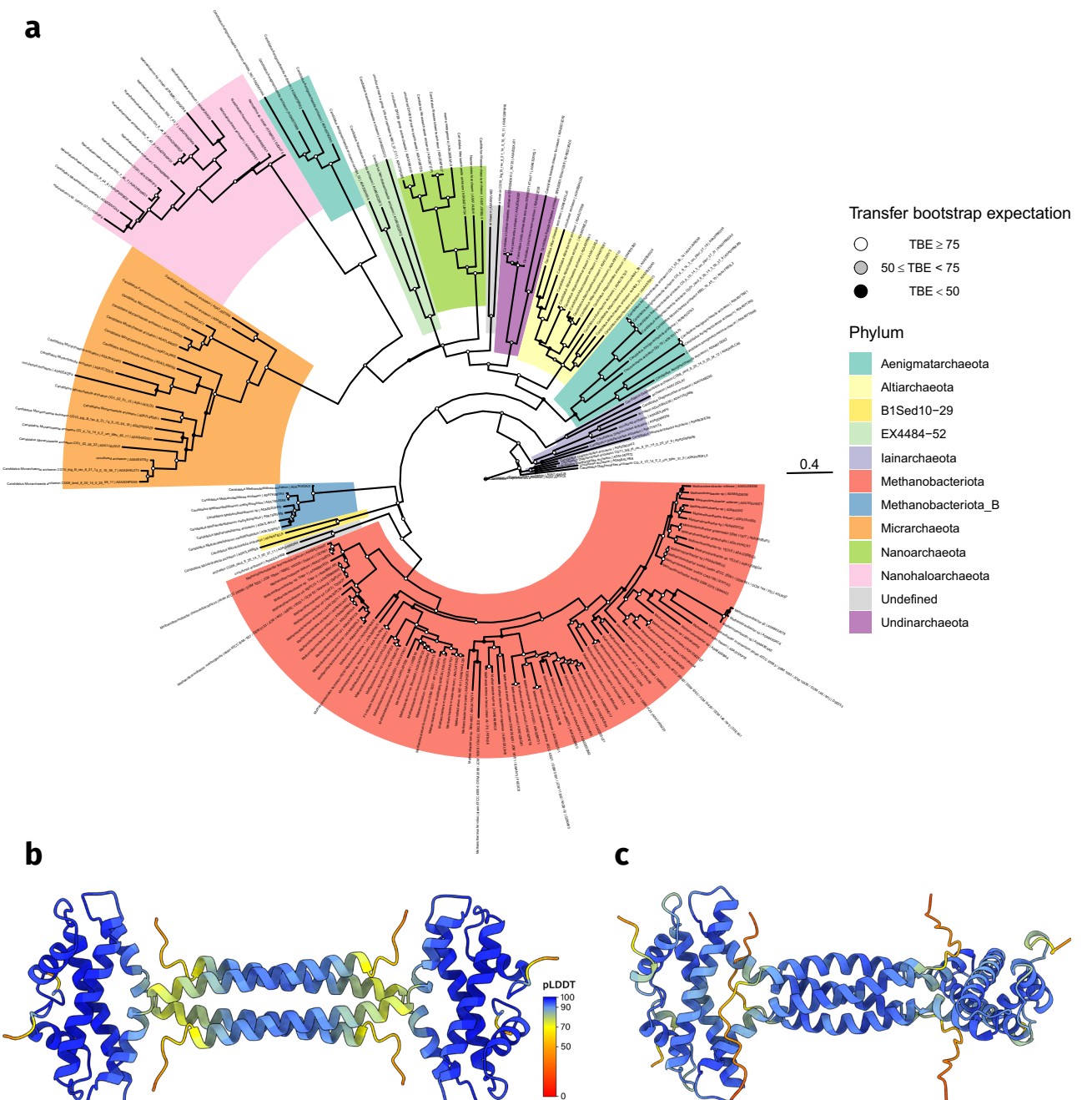

**Fig. 6 | DNA-bridging coiled-coil (CC) histones are widely found in archaea.**
**a** Phylogenetic tree of CC histones. Clades are colored by phylum as they are assigned in the GTDB database (v207). The tree was generated with RAxML-NG. 1000 bootstraps were performed and used to calculate the transfer bootstrap expectation values (TBE). **b** The homotetramer of CC histone E3GZL0 from *Methanothermus fervidus* as predicted by AlphaFold2. CC histones contain a long α-helix on the C-terminus. Each residue is colored by its pLDDT value. **c** The homotetramer of RdgC histone D4GVY1 from *Haloferax volcanii* as predicted by AlphaFold2. RdgC histones form very similar tetramer structures to coiled-coil histones despite low sequence identity. Each residue is colored by its pLDDT value.

These are the Methanococcales, coiled-coil, RgdC, and the aforementioned phage histones. The Methanococcales (Mc) histones are exclusively found in the Methanococcales order and contain a tetramerization domain on the C-terminus which facilitates DNA bridging. This has been experimentally confirmed for the Mc histone MJ1647 from *Methanocaldococcus jannaschii*[19]. Coiled-coil (CC) histones are more widely distributed throughout archaea, being found in Aenigmatarchaeota, Altiarchaeota, B1Sed10-29, EX4484-52, Iainarchaeota, Methanobacteriota(_B), Micrarchaeota, Nanoarchaeota, Nanohaloarchaeota, and Undinarchaeota (Fig. 6a). CC histones have a long α-helix at the C-terminus of the histone fold, which is predicted to form a coiled-coil in the tetramer (Fig. 6b). The tetramer structure of CC histones bears structural resemblance to that of Mc histones, where the two dimers are positioned opposite each other and interact through their C-terminal domains. However, despite these structural parallels, there is little sequence similarity between CC and Mc histones. Based on the HMM logo, we identified 4 possible DNA binding residues on the α1 helix face of CC histones: R32, K35, R45, and K49 (Supplementary Fig. 24). Two of these, R32 and K35, correspond to the DNA binding residues R20 and K23 in the nucleosomal HMM logo (Supplementary Fig. 10). A common feature of CC histones is their highly negatively charged tails (Supplementary Figs. 26 and 27). These

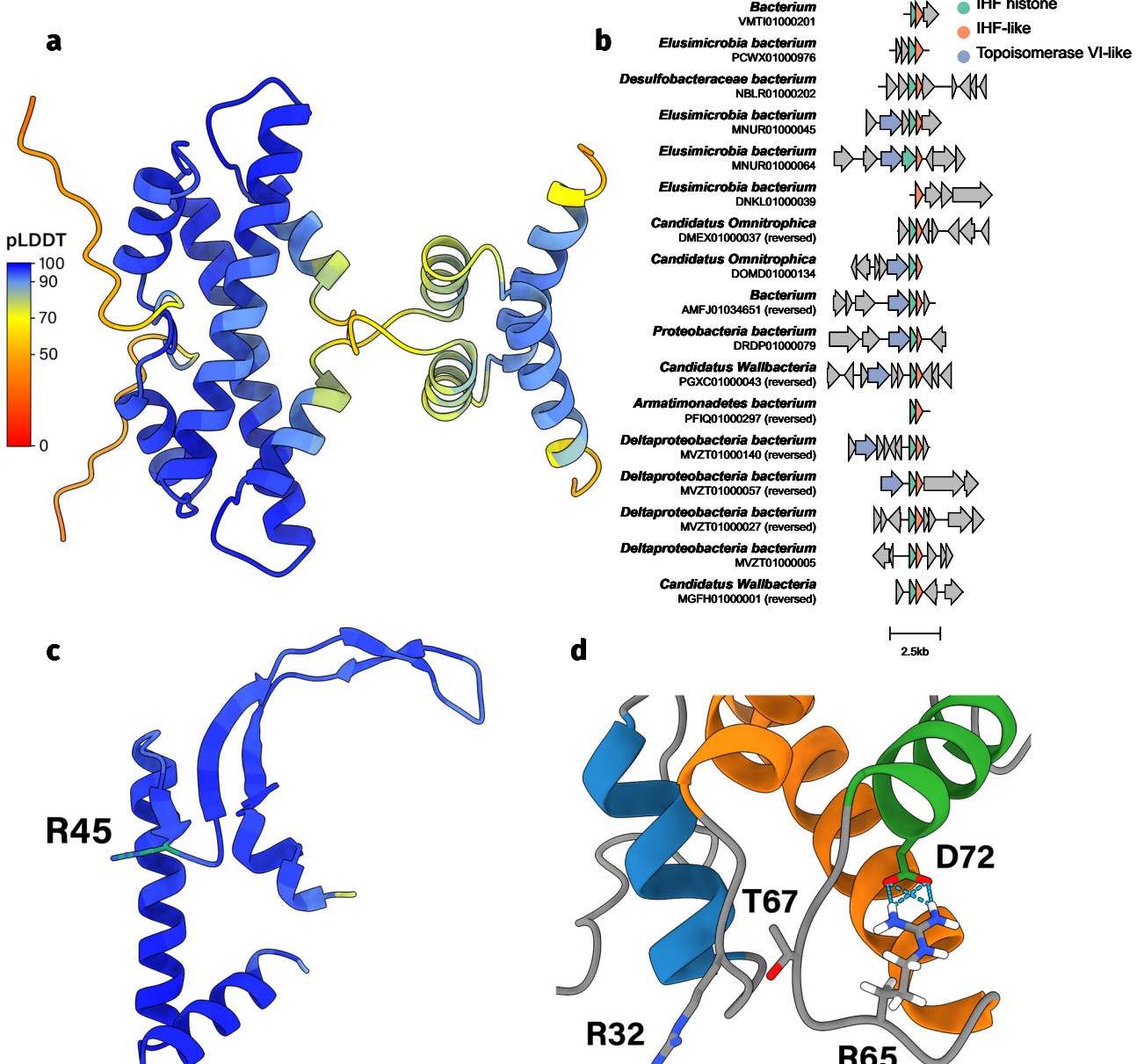

**Fig. 7 | IHF histones are functionally related to IHF. a** The homodimer of IHF histone A0A358AGI2 from *Candidatus Omnitrophica* as predicted by AlphaFold2. Each residue is colored by its pLDDT value. **b** Gene cluster comparison of bacterial metagenomes that contain the IHF histone. The organism and its genome ID are noted on the left. The IHF histone, IHF-like, and topoisomerase VI-like genes are colored green, orange, and blue respectively. **c** The monomer of IHF-like tails can be present at the C- and/or N-terminus and are predicted by AlphaFold to be disordered. Notably, the CC histone of model archaeon *Methanothermus fervidus* lacks these tails. The function of the tails remains unclear; however, as they are highly negatively charged, they might act as intramolecular inhibitors by occluding the DNA-binding α1 face of the histone. Transcriptome data show that CC histone of *Methanobrevibacter smithii* PS shows low expression, ranking in the top 53% of the highest expressed genes (Supplementary Fig. 28).

The predicted tetramer structure of CC histones suggests that they bridge DNA by binding two separate DNA duplexes at opposing histone dimers (Supplementary Fig. 25a). To test this hypothesis, we purified the CC histone from *M. fervidus*, which we refer to as HMfC, and performed a DNA-bridging assay (Supplementary Fig. 25b). We

A0A358AGI6 from *Candidatus Omnitrophica* as predicted by AlphaFold2. Each residue is colored by its pLDDT value. Residue R45 is highlighted in green. **d** The RxTxxxxD motif and the "sprocket" R32 of IHF histone A0A358AGI2. R32, R65, T67, and D72 relate to R24, R57, T59, and D64 in the IHF histone HMM logo (Supplementary Fig. 34).

observe an increase in DNA bridging activity with increasing HMfC concentrations, confirming that CC histones bridge DNA. This represents a major divergence from the DNA binding mode of conventional histones and highlights both the diversity of prokaryotic histones and the utility of AlphaFold in providing accurate preliminary insights into the DNA-binding properties of histones.

RgdC histones are similar to CC histones in that they contain a large C-terminal helix which forms a tetrameric coiled-coil helical bundle (Fig. 6c). However, they share only 15 to 20% sequence identity with CC histones. RdgC histones are present in certain Bacillus and Halobacteria species and are encoded within multi-gene operons. The gene for the RdgC histone is always the first gene in its operon, followed by an RdgC-like protein and an unkown transmembrane (TM) protein (Supplementary Fig. 29). The conservation of this operon

structure across different organisms suggests functional coupling among these proteins. The RdgC-like protein is structurally similar to RdgC, yet they share only 18% sequence identity (Supplementary Fig. 30). RdgC is found in Proteobacteria and forms a ring structure as a dimer, through which it might bind DNA[41]. Functionally, it is involved in recombination and is thought to modulate RecA activity[42]. The RdgC-like protein differs from RdgC as it contains two additional small domains, one on the C-terminus and the other on the N-terminus. The TM protein is found exclusively within the context of the RdgC histone. It consists of 4 domains: a winged helix domain, two unknown domains, and a transmembrane domain (Supplementary Fig. 31). Identifying functional residues in RdgC histones is challenging due to significant variability in sequences across species (Supplementary Fig. 32). One of the few conserved DNA binding residues is K34, which corresponds to the DNA binding residue K23 in the nucleosomal HMM logo (Supplementary Fig. 10). Although the sequence variation is high, the $\alpha1$ face is always positively charged, suggesting they all bind DNA. Transcriptome data from *H. volcanii* and *Bacillus cereus* strain ATCC 10987 show low expression of the RdgC histone, ranking it within the top 14 to 37% of highest expressed genes (Supplementary Figs. 12a, b, 33).

### IHF-related histones

Genes encoding nucleoid-associated proteins are generally organized as single-gene operons[43]. This pattern holds for model histones HMfA, HMfB, HTkA, and HTkB, as well as for the previously discussed $\alpha3$ and CC histones. However, as described earlier, some histones consistently occur within multi-gene operons across different organisms. One such example is what we refer to as IHF-related or IHF histones (Fig. 7a). IHF histones are scarcely found in bacterial metagenomes, with only 19 IHF histone homologs in the UniProt database. Characterized by a structured C-terminal tail that dimerizes, IHF histones likely function as dimers as AlphaFold does not confidently predict larger homo-oligomer structures. They appear in metagenomes from the phyla Omnitrophota, Wallbacteria, CG03, and Elusimicrobiota. In all cases, a gene encoding an integration host factor-like (IHF-like) protein is present within the same operon as the genes for IHF histones, hence the name 'IHF histones' (Fig. 7b and c). The conserved co-occurrence of these proteins suggests that they are associated via an unknown function.

Both IHF histones and IHF-like proteins exhibit low sequence similarity across organisms. The most strongly conserved residues in the histone fold are the RxTxxxxD motif and "the sprocket" R24 (Fig. 7d and Supplementary Fig. 34). The typical residues in the $\alpha1$ helix facilitating DNA binding are not conserved, except the hydrophobic residues involved in the packing of the dimer. Thus, it remains unclear whether IHF histones bind DNA similarly to nucleosomal histones. The C-tail contains two $\alpha$-helices that dimerize in a handshake motif. As the tail lacks strongly conserved positively charged residues, it likely does not bind to DNA. The IHF-like protein shows weak sequence similarity to both HU$\alpha/\beta$ and IHF$\alpha/\beta$. However, the IHF-like protein might function similarly to IHF$\beta$ based on the strong conservation of R45, a residue conserved in IHF$\beta$ but not in HU or IHF$\alpha$ (Supplementary Fig. 35)[44]; Furthermore, we propose that the IHF-like protein can bend DNA, similar to IHF, based on the conservation of intercalating and DNA binding residues on the beta arms (Supplementary Fig. 36).

## Discussion

We have discovered a prokaryotic histone landscape far more diverse than previously recognized. We categorized all prokaryotic histones into 17 distinct groups based on their predicted AlphaFold structures. All histones exhibit the conventional histone fold, however they differ in their predicted quaternary structures. Notably, we identified a major histone family, the $\alpha3$ histones, which are prevalent across nearly all archaeal phyla, several bacteria taxa, and some bacteriophages. For face-to-face (FtF) histones, which form the largest subgroup of $\alpha3$ histones, we confirmed their unique quaternary structure by crystallography. Furthermore, we have identified histones throughout archaea that exhibit DNA bridging properties, a major divergence from the DNA wrapping properties of nucleosomal histones.

We hypothesize that $\alpha3$ histones likely originated in archaea, given their more prevalent occurrence within the archaeal domain compared to the bacterial domain. Furthermore, the conservation of the nucleosomal RxTxxxxD motif suggests that $\alpha3$ histones and nucleosomal histones may share a common evolutionary ancestor. Bacteria might have acquired FtF histones through horizontal gene transfer. Our phylogenetic tree of $\alpha3$ histones shows several examples where horizontal gene transfer likely occurred between archaea and bacteria, such as the common ancestor histone between Leptospira and Iainarchaeota FtF histones. The bacteria specific $\alpha3$ histones, such as the bacterial dimer and ZZ histones, might have evolved from these horizontally acquired FtF histones. Furthermore, given the sporadic distribution of $\alpha3$ histones across diverse bacterial lineages, another explanation, which cannot be entirely dismissed, is that $\alpha3$ histones may trace back to the last universal common ancestor (LUCA). This hypothesis suggests that these histones were initially more universally distributed but did not significantly diversify in bacteria or were largely lost, remaining only sporadically across different lineages.

The origins of some histone groups remain elusive. The IHF histones, for instance, are found in only 19 metagenomes, spanning a diverse range of organisms. We refer to these rare histones as "exotic" histones as they do not seem to be conserved by any known group of organisms. Typically, proteins not conserved across any group of organisms tend to disappear over evolutionary timeframes. However, it is possible that these exotic histones are preserved in yet-to-be-discovered organisms.

Determining the functions of histones is challenging from their structure alone. To infer potential function, we have analyzed the predicted structures and conserved residues, comparing them to existing literature. For instance, the predicted tetrameric structure of coiled-coil histones closely resembles the structure of the bridging histone MJ1647, leading us to hypothesize that coiled-coil histones may also bridge DNA. However, the literature on prokaryotic histones is currently limited. We hope that our findings will serve as a valuable resource for the scientific community, encouraging further experimental investigation of these histones. In the case of coiled-coil histone HMfC, we show that it can bridge DNA, however, the structure of this DNA-bridging complex is still unclear. We hypothesize that HMfC bridges two separate DNA duplexes as a tetramer, where two dimers interact through their C-terminal extensions, forming a coiled-coil tetramer. Therefore, we expect that removing the C-terminal extension or weakening the interactions within the coiled-coil domain will result in a loss of tetramerization and, thus, DNA bridging. This mutant will only bind DNA as a dimer and might function similarly to the bacterial histone HBb, which bends DNA as a dimer[26]. We hope that future studies can clarify how HMfC binds and bridges DNA. One of the main experimental hurdles is that several histone groups are either found exclusively in metagenomes or in archaea which are notoriously difficult to culture. Therefore, their in vivo roles can be difficult to ascertain. The most readily accessible groups are the FtF, coiled-coil, RdgC, bacterial dimer, and ZZ histones as these can be found in culturable bacteria or archaea. Furthermore, although we refer to our histone-fold proteins as "histones", it remains uncertain whether they all perform the conventional function of histones as global genome organizers. Similar to how TAF proteins have evolved from the eukaryotic core histones, some histones could have evolved to gain functions not related to genome organization. For example, histones within the DUF1931 and the transmembrane categories lack conventional histone DNA-binding residues, suggesting potential roles beyond genome organization.

Lastly, while our bioinformatical analysis focuses solely on histones, the methodology employed, particularly the use of AlphaFold2 to find remote homologs and to gain insight into their function, is applicable to all proteins. This broad application is possible because almost all UniProt entries now feature AlphaFold predictions. While traditional structural alignment programs are too slow to efficiently search through this massive amount of structural data, the recent development of FoldSeek allows anyone to find structurally similar proteins within a few seconds[45]. These structural analyses will bring new insights into the conservation of proteins across life and elucidate the relationships between structure, function, and sequence.

## Methods

### Identifying and predicting histones

To generate a list of histones in prokaryotes based on sequence, all bacterial, archaeal, and viral histone-fold proteins in InterPro (IPR009072) were retrieved[30]. Multiple sequence alignments (MSAs) were generated with MMseqs2 at a search-sensitive parameter of 8[46]. Target databases used for the MSAs were constructed by the Colab-Fold team (https://colabfold.mmseqs.com/) and include UniRef30, BFD, Mgnify, MetaEuk, SMAG, TOPAZ, MGV, GPD, and MetaClust2[47]. The constructed MSAs were used as an input for LocalColabFold to predict the protein structures. Monomer, dimer, tetramer, and hexamer structures were predicted for all histones with AlphaFold2 and AlphaFold2-Multimer (v2)[27,28]. No templates were used as templates generally do not improve the quality of predictions[27]. We only ran homomeric predictions as we did not have the computing power to predict all possible heteromeric complexes in a reasonable timeframe. The monomer, dimer, and tetramer predictions were done with 6 recycles and the structures were relaxed by AlphaFold's AMBER forcefield. The hexamer predictions were done with 3 recycles and no relaxation to save computation time. The dimer prediction for A0A0F8XJF6 and the tetramer prediction for A0A2E7QIQ9 were done with 12 and 32 recycles respectively. Both MMseqs2 and LocalColab-Fold were run on the high-performance computing facility ALICE at Leiden University. The predicted structures of all histones shown in this article are available in the Supplementary Data.

### Histone classification

All histones were manually classified based on their predicted structures, meaning we inspected every rank 1 prediction in ChimeraX, the pLDDT values, and the rank 1 PAE plot. Classification was done by examining all the predicted monomer and multimer structures, the predicted alignment errors (PAE), and the predicted local distance difference test (pLDDT). Each category consists of histones that form similar unique multimer structures or which contain additional domains fused to the histone domain. We examined the predicted alignment error for each multimer prediction to assess AlphaFold's confidence in the multimer interface. Interfaces with PAE values < 10 Å were considered high quality. Low-quality interfaces were not considered for categorization. Proteins that could not be classified based on their predicted structures, i.e., histones with only confident monomer and dimer predictions and no defining features in the monomer structure, were classified as "undefined". The only exceptions are histones we classified as "bacterial dimers", which are sequentially highly related to each other and are exclusive to bacteria. We identified these bacterial dimer histones from the CLANS clustering. They were initially classified as "undefined" as they contain no unique quaternary structural features. However, these bacterial dimer histones clustered close together in CLANS. We, therefore, decided to group them into a category. The bacterial dimer category is the only category that was defined based on sequence clustering instead of on predicted structure. Histones that contained eukaryotic sequences in their multiple sequence alignments with a sequence identity above

60% were classified as "eukaryotic-like" and likely represent contaminations.

### CLANS clustering

CLANS clustering was performed with the MPI Bioinformatics Toolkit and the CLANS java application[32,48]. Briefly, the online CLANS toolkit performed a pairwise sequence alignment of all histones. This sequence similarity matrix was loaded in CLANS (java application) and clustering was performed for 16407 rounds. Default parameters were used except for the minimal attraction parameter, which was set to 50. From CLANS, the graph data and the attraction values between vertices were exported. The clustering was subsequently visualized in Python 3 with matplotlib. The histone categories, as determined based on the predicted structures, were visualized by generating alpha shapes.

### Hidden Markov Profiles (HMM)

For each histone category, an HMM profile was generated. For each category, all histones were aligned in a multiple sequence alignment using Muscle v5 with the Super5 algorithm[49]. From the multiple sequence alignments, an HMM profile was generated with HMMER (v3.3.2)[50]. With Skylign a logo was generated for each HMM profile[51]. The HMM profiles were also used to search against UniProt to find additional histones that were not annotated by InterPro. All HMM profiles are available in the Supplementary Data.

### Phylogenetic trees

The coiled-coil and α3 histone phylogenetic trees were generated with RAxML Next Generation (v1.1.0)[52]. Multiple sequence alignments were made with the coiled-coil and α3 histone nucleotide sequences using Muscle v5 (Super5 algorithm)[49]. In the case of the α3 histone phylogenetic tree, the alignment was truncated to remove the nucleotides corresponding to the GTPase and ZZ domains. For both trees, a GTR model with 4 free rates was used with a maximum likelihood estimate of the base frequencies (GTR+R4+FO). For the α3 tree, 40 tree searches were performed using 20 random and 20 parsimony starting trees. For the coiled-coil tree, 60 tree searches were performed using 30 random and 30 parsimony starting trees. 1000 and 600 bootstraps were performed for the coiled-coil and the α3 trees respectively. The bootstrap trees were used to compute the Transfer Bootstrap Expectation (TBE) support metric. The trees were visualized in R with the ggtree package[53]. To determine which histone belongs to which phylum in the GTDB database, the GenBank assembly accession of each histone was searched against the GTDB database (v207)[54].

### Cladogram

The cladogram of the archaeal and the bacterial superkingdoms were generated in R with the ggtree package[53]. The tree data from GTDB version 207 was used[54]. To determine which histone belongs to which bacterial phylum in the GTDB database, the GenBank assembly accession of each histone was searched against the GTDB database (v207).

### Structural similarity search

For the identification of unknown domains or proteins, FoldSeek was used[45]. The structure data of the domains in question, as predicted by AlphaFold, were used as inputs for FoldSeek. They were searched against FoldSeek's PDB100 and the AlphaFold Swiss-Prot databases using the webserver's default parameter. Structural similar proteins were subsequently analyzed in ChimeraX.

### Gene clusters

For gene clustering, Clinker was used[55]. For each histone, the genomic context, 5000 nucleotides before and after the histone gene, was retrieved from the NCBI database using Entrez. These genome files

were used as inputs for the Clinker web server using default parameters. Briefly, Clinker performs an all vs all global alignment between genes and automatically detects gene clusters based on sequence similarity. The resulting alignment and gene clusters were manually analyzed and genes were classified based on sequence and predicted AlphaFold structures.

## Cloning, expression, and purification of face-to-face histone from *Thermococcus kodakarensis* (HTkC)

The gene *TK1040* was ordered as a synthesized product (GeneArt; Thermo Fisher Scientific) and was cloned into a pET30b expression vector through Gibson Assembly. The sequences of the oligonucleotides are available in the Supplementary Information. The sequence of the construct was verified by DNA Sanger sequencing (BaseClear). The plasmid (pRD503) was deposited at Addgene (Supplementary Table 4). pRD503 was transformed into chemically competent BL21(DE3)pLysS cells (Novagen) through heat shock. Lysogenic broth (LB) supplemented with 50 µg mL$^{-1}$ kanamycin and 25 µg mL$^{-1}$ chloramphenicol was inoculated with 1% (v/v) of a starter culture of transformed cells grown overnight. The culture was grown at 37 °C with 200 RPM shaking, and upon reaching an OD600 of approximately 0.6, IPTG was added to a final concentration of 1 mM to induce expression. After growing the cells further for 1 h, the culture was harvested at 7510 × $g$ for 30 min at 4 °C. The cell pellet was resuspended in 50 mM Tris-HCl, pH 8, 150 mM NaCl, 10 mM MgCl$_2$, and stored at −70 °C. After thawing the cell pellet, PMSF and DNase were added, and lysis was done with a Stansted Pressure Cell Homogenizer S-PCH-10 (Homogenizing Systems Ltd.) at 310 MPa. The lysate was centrifuged at 16000 xg for 30 min at 4 °C. The resulting supernatant was then heated at 70 °C for 15 min, followed by another round of centrifugation at 16,000 × $g$ for 30 min at 4 °C. The supernatant was run on a pre-equilibrated 5 mL HiTrap Heparin HP (Cytiva) attached to an NGC Chromatography System (BioRad). A 50 mL gradient of 50 mM Tris-HCl, pH 8, 100 mM KCl to 50 mM Tris-HCl, pH 8, 1 M KCl was applied, and the eluent was fractionated into 1 mL fractions. After SDS-PAGE verification, the fractions containing the protein were pooled and concentrated using Vivaspin® Turbo 4 3000 MWCO centrifugal concentrators. The concentrated sample was applied to a Superdex 75 Increase 10/300 GL size exclusion column equilibrated with 100 mM KCl, 50 mM Tris-HCl, pH 7, 10% glycerol. The protein present in the collected fractions corresponding to the peak was then characterized with SDS-PAGE, intact protein LC-MS, and circular dichroism spectroscopy, and stored at −70 °C.

## Crystallization, data collection, and structure determination of HTkC

For crystallization, HTkC was concentrated to 4.0 mg mL$^{-1}$ in 50 mM Tris-HCl, pH 7.0, 100 mM KCl and 10% glycerol. Crystallization trials were performed using commercial crystallization screens in 96-well sitting drop vapor-diffusion plates with a reservoir volume of 100 µL and droplets consisting of 300 nL of protein and 300 nL reservoir solution. The best-diffracting crystals grew with a reservoir solution containing 0.01 M trisodium citrate and 33% (w/v) PEG 6000 within 1–3 days. Before loop-mounting and flash-cooling in liquid nitrogen, the crystals were transferred to a droplet of the reservoir solution. Data were collected at beamline X10SA of the Swiss Light Source (Villigen, Switzerland) at 100 K, using an EIGER X 16M hybrid pixel detector (Dectris Ltd., Switzerland). The data were reduced, processed, and scaled in space group P2$_1$2$_1$2$_1$ to a resolution of 1.84 Å using XDS[56]. The structure was solved by molecular replacement with MOLREP[57] using an AlphaFold[27,28,47] prediction as a model, locating four monomers in the asymmetric unit. The structure was refined in cycles by manual modeling in Coot[58] and refinement with REFMAC5[59]. Data processing and refinement statistics are given in Supplementary Table 1. The coordinates and structure factors are deposited in the PDB under accession number 9F2C.

## Cloning, expression, and purification of coiled-coil histone from *Methanothermus fervidus* (HMfC)

The gene *Mfer0945* was ordered as a synthesized product (GeneArt; Thermo Fisher Scientific) and was cloned into a pET30b expression vector through Gibson Assembly. The sequences of the oligonucleotides are available in the Supplementary Information. The sequence of the construct was verified by DNA Sanger sequencing (BaseClear). The plasmid (pRD551) was deposited at Addgene (Supplementary Table 4). pRD551 was transformed into chemically competent BL21(DE3)pLysS cells (Novagen) through heat shock. Lysogenic broth (LB) supplemented with 50 µg mL$^{-1}$ kanamycin and 25 µg mL$^{-1}$ chloramphenicol was inoculated with 1% (v/v) of a starter culture of transformed cells grown overnight. The culture was grown at 37 °C with 200 RPM shaking, and upon reaching an OD600 of approximately 0.5, IPTG was added to a final concentration of 1 mM to induce expression. After growing the cells further for 3 h, the culture was harvested at 7510 × $g$ for 30 min at 4 °C. The cell pellet was resuspended in 25 mM Tris-HCl, pH 8, 1.5 M NaCl, 10% glycerol, and stored at −70 °C. After thawing the cell pellet, PMSF and DNase were added, and lysis was done with a Stansted Pressure Cell Homogenizer S-PCH-10 (Homogenizing Systems Ltd.) at 310 MPa. The lysate was centrifuged at 16,000 × $g$ for 30 min at 4 °C. The resulting supernatant was then heated at 70 °C for 15 min, followed by another round of centrifugation at 16,000 × $g$ for 30 min at 4 °C. The supernatant was filtered (0.2 µm syringe filter) and diluted four times with binding buffer (150 mM NaCl, 25 mM Tris-HCl, pH 8, 10% glycerol). This mixture was run on a pre-equilibrated 5 mL HiTrap Heparin HP (Cytiva) attached to an NGC Chromatography System (BioRad). A 50 mL gradient of 150 mM to 1.5 M NaCl in binding buffer was applied, and the eluent was fractionated into 1 mL fractions. After SDS-PAGE verification, the fractions containing the protein were pooled and heated at 70 °C for 15 min. The mixture was centrifuged at 16,000 × $g$ for 30 min at 4 °C, and the supernatant was diluted four times with binding buffer and subjected to another round of heparin chromatography. The peak fractions were pooled and applied to a HiLoad 10/600 Superdex 75 pg size exclusion column equilibrated with 300 mM KCl, 25 mM Tris-HCl, pH 8, 10% glycerol. The protein present in the collected fractions corresponding to the peak was then characterized with SDS-PAGE, intact protein LC-MS, and circular dichroism spectroscopy, and stored at −70 °C.

## DNA-bridging assay with HMfC

The DNA-bridging assay was performed using a 47% GC 685 bp DNA substrate as previously described[19,60]. In short, the DNA substrates were generated by PCR using Thermo Scientific® Phusion® High-Fidelity DNA Polymerase for prey DNA and with a biotinylated primer for bait DNA. The subsequent PCR products were purified using a GeneElute PCR Clean-up kit (Sigma Aldrich) and prey DNA was 32P labeled with Polynucleotide Kinase and γ-32P ATP. For each bridging assay measurement, 6 µL of streptavidin-coated paramagnetic beads were washed with 50 µL PBS (12 mM NaPO4 pH 7.4, 137 mM NaCl). The beads were subsequently washed twice with 50 µL CB (20 mM Tris-HCl pH 8.0, 2 mM EDTA, 2 M NaCl, 2 mg mL$^{-1}$ Acetylated BSA, 0.04% Tween20) and then resuspended in 6 µL of CB. Next, 100 pM (in a total volume of 3 µL) of biotinylated bait DNA was added to half of the suspension and incubated with the beads at 25 °C for 20 min in an Eppendorf Thermomixer with an Eppendorf Smartblock 1.5 mL at 1000 rpm. The other half of the bead suspension was incubated without DNA as a control. After incubation, the beads were washed twice with 16 µL incubation buffer and resuspended in 16 µL incubation buffer. 2 µL of protein and 2 µL of radiolabeled prey DNA probes (with a minimum of 5000 counts per minute) were added to both bead suspensions, gently mixed and incubated at 25 °C for 20 min in an Eppendorf Thermomixer with an Eppendorf Smartblock 1.5 mL at 1000 RPM. Incubation buffer, DNA buffer, and protein buffer were designed in such a way to make a constant experimental buffer: 10 mM Tris-HCl, pH 7.0, 51 mM KCl, 5% v/v glycerol, 1 mM spermidine, 1 mM DTT, 0.02%

Tween20, 1 mg mL$^{-1}$ acetylated BSA, 1 mM MgCl2. After incubation, the beads were gently washed with 20 µL of the same experimental buffer and then resuspended in stop buffer (10 mM Tris-HCl pH 8.0, 1 mM EDTA, 200 mM NaCl, 0.2% SDS). The sample was transferred to a liquid Cherenkov-scintillation counter to quantify the radioactive signal. The calculation of DNA recovery (%) was based on a reference sample containing the same amount of labeled 32P 685bp DNA used in each sample. All measurements were performed at least in triplicate.

### Reporting summary
Further information on research design is available in the Nature Portfolio Reporting Summary linked to this article.

## Data availability
The list of prokaryotic histones, the predicted AlphaFold structures shown in this article, the HMM profiles, and the DNA-bridging data are available from the 4TU repository (https://data.4tu.nl) at https://doi.org/10.4121/d268a6a9-2fc5-46aa-a236-0ca3d3f7ed75. X-ray structural data was deposited in the PDB database with accession number 9F2C. Source data are provided with this paper.

## Code availability
Scripts required to reproduce figures are available at https://github.com/SamuelSchwab/Histones-and-histone-variant-families-in-prokaryotes.

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

## Acknowledgements

Finn Werner and Fabian Blombach are kindly acknowledged for thorough reading and commenting on the manuscript. The ALICE HPC cluster at Leiden University is kindly acknowledged for providing the infrastructure necessary to perform many of the computations described in this article. UCSF ChimeraX is kindly acknowledged for the molecular graphics and analyses in this article. We are grateful to the staff of Beamline X10SA of the Swiss Light Source (PSI, Villigen, Switzerland) for excellent technical support. We extend our thanks to Reinhard Albrecht for assistance with crystallization and crystallographic data collection. This work was funded by the Netherlands Organization for Scientific Research [OCENW.GROOT.2019.012 to R.T.D.].

## Author contributions

Samuel Schwab performed all computational analyses described in this article, wrote the article, and purified HTkC. Yimin Hu and Marcus D. Hartmann crystallized HTkC and solved the crystal structure. Bert van Erp and Marc K. M. Cajili purified HMfC and performed the bridging assay. Birte Hernandez Alvarez and Vikram Alva reviewed and corrected the manuscript. Remus T. Dame and Aimee L. Boyle supervised the project and reviewed and corrected the manuscript.

## Competing interests

The authors declare no competing interests.
