## [Peer Review File · Nature Communications]

Histones and histone variant families in prokaryotesReviewers' Comments:

Reviewer #1:

Remarks to the Author:

This manuscript uses AlphaFold2 predictions and other bioinformatic tools to characterize all detectable histone proteins in prokaryotes. 5800 histones (as annotated by InterPro) are identified and their structures predicted by AlphaFold2. The remainder of this manuscript goes through all 17 families present in this data, and -- using predicted structure and sequence for each of these families -- hypothesizes on the overall subunit architecture, DNA-binding interface, and overall mode of DNA organization.

While the study is very thorough, and the data is very clearly presented, I feel this work is only of interest to a very specialized audience. I also have some concerns about the overall value of these structures for analysis, particularly as AlphaFold is unable to predict protein/DNA interfaces, and is only moderately accurate at inferring higher-order structure. Some specific comments are listed below:

- it is not completely clear when structure and when sequence is used for classification. The text says that new histones are "categorized" by structure, but the methods section (and figures) only describe sequence-based clustering. This needs to be more clearly specified in the text.

- more generally, it is difficult to tell what structure prediction is adding to the analysis. Which of the families from Figure 2a already have at least 1 crystal structure determined? For those cases, wouldn't homology modeling give the same level of detail of analysis (topology and putative binding residues & architecture)? While AlphaFold2 certainly gives more accurate predictions, it is not clear what that increased accuracy is adding to the overall conclusions. Are there specific examples where predicted structure was necessary for analysis? Or corrected something misclassified in the sequence-based clustering?

- More information should be provided on the AlphaFold modeling. Of the 5800, how many had confident predictions? Were MSAs deep for all of them? Why were templates not used? (Did MSAs alone give very high-confidence predictions?) Were heteromeric predictions run at all? Are any of the histones believed to be heteromeric?

Reviewer #2:

Remarks to the Author:

The manuscript by Schwab et al. identifies histone fold proteins in prokaryotes and uses AlphaFold 2 to classify them into 17 different structural groups, 13 of which are previously undescribed. The authors analyze conserved residues in the various groups, try to predict whether and how they bind DNA, look at their phylogenomic distributions, and examine expression data to determine the abundance of these proteins, many of which are highly expressed. This paper greatly expands our knowledge of histone fold domain proteins in archaea and bacteria, and should be a spur to further investigation of this important protein family. My comments are mostly minor.

Figure 2a, Do the authors have any comment or explanation of why TM and RdgC have disjunct distributions? Are the RdgC-like proteins in this figure?

Page 4, bottom, "17 categories, 13 of which are new". This would be a good place to reference Table S1. In the Table, the authors could also mark the four previously known categories (maybe in bold font). For DUF1931 and Methanococcales they mentioned a known example from the class under discussion. They could do likewise for the bacterial dimer and nucleosomal categories.

Page 5, top, the first mention of Figure S2 that I can find comes in the fold-to-fold section, after mentioning Figure S4. Did I miss something?

Page 5, bottom, the numbering of amino acids in the text does not match the numbers in Figure 3, which is very confusing. In the legend to Figure 3 the correspondence between the numbers in the model of D4GZE0 and the Logos of the HMM model is listed, but is there a better way to do this? For example, after saying on page 5 that the conserved residues in the HMM are 48, 52, 54, 56, and 61, can the authors say these correspond to N45, R47, T49, and D54 in D4GZE0 and then use the latter numbers that match the figure?

Also in Figure 3 Legend, please explain what pLDDT values are. The acronym is explained in the Methods, but people will read the figure legends first before they get to the Methods.

In Figure 4 the amino acid numbers in the legend again do not match the figure and are off by 10. Please refer to the numbers that match the figure, and then you can mention the corresponding numbers in the logo in Figure S13 at the end of the legend.

In the legend to Figure S14 I suggest listing the organism after each protein ("D0LYZ1 from *Haliangium ochraceum* SMP-2 and E1WYC3 from *Halobacteriovorax marinus* SJ"). I was initially slightly confused by the way it is currently listed.

Page 8, end of second paragraph states that there is low expression of ZZ histone in both attack and growth phase, citing Figure S11; however, I only see ZZ marked in the attack phase in Figure S11b. Is there evidence for its abundance in the growth phase? Can it be marked in Figure S11a?

Page 9, it is unclear why the conserved DNA-binding residues of bacteriophage histones are listed as R10 and R49 when the logo in Figure S17 implies K is much more common at positions 10 and 49.

The authors' focus is on poorly known prokaryotic histones, but I wonder if in their discussion they want to compare to non-nucleosomal eukaryotic histones such as the TAFs with histone fold domains, at least some of which lack DNA-binding residues. A somewhat similar analysis of proteins with histone fold domains was recently published for plants (PMID: 33888032), with diverse functions for histone folds that might provide additional context for their findings.

Reviewer #3:

Remarks to the Author:

The DNA binding proteins in bacteria are typically histone-like but are more diversely defined by their functions compared to that in eukaryotes. This work used the deep-learning based protein structure prediction tool AlphaFold2 to predict the monomer and multimer structures of prokaryotic histone-fold proteins (searched from histone-fold homologous superfamily). The authors analyzed the predicted structures, categorizing them into 17 distinct groups. They found that along with nucleosomal histones, which are typical in eukaryotes, prokaryotes contain a second major histone family, where a shorter $\alpha 3$ helix was found, termed as $\alpha 3$ histone. The authors then discussed several interesting biological applications. Overall, it is a nice bioinformatics study of histone proteins with some interesting findings. However, I am afraid that I am not persuaded that the degree and solidness of the advance provided, and the novelty of their findings are sufficient to match the high standard of Nature Communications.

My major concern is that the AlphaFold2 (AF2) method, which is deeply trained on the PDB structures and typically uses template search for the best performance, may have structural bias in its predictions. Without experimental or further verification, the biological stories out of it are not convincing. Also, since the query sequences that the authors used are from the result of homologous

sequence search of histone-fold, it is not surprising that AF2 would predict the main histone-fold structure with shorter or longer local secondary structures at different places. Further, most of the predictions done in this work are among multiple histones as we know that histone monomer does not have stable structure. The AF2-complex is known to not have comparable performance vs. the AF2-monomer. Specific setups or processing are needed to enhance a good performance as has been shown in the recent CASP15 test (<https://doi.org/10.1016/j.sbi.2023.102594>). pIDDT is a metric used in AF2 to describe the prediction confidence of the local regions. As another observation from the predicted structures, most of the interesting parts that support the authors' points seem to have lower pIDDT than the main histone-fold frame.

Reviewer 1

>It is not completely clear when structure and when sequence is used for classification. The text says that new histones are "categorized" by structure, but the methods section (and figures) only describe sequence-based clustering. This needs to be more clearly specified in the text.

This is a good point. Almost all of the categorisation was done manually based on the predicted structures. We go through every prediction manually by viewing every rank 1 prediction in ChimeraX and we assess the quality of each prediction based on the predicted aligned error (PAE) and pLDDT values. Histones that are predicted to form similar unique quaternary structures are categorised together if the inter-chain interface has low distance errors ($<10\text{\AA}$) in the PAE plot. Furthermore, we also categorise histones together if they have additional domains, such as the ZZ histones which all have a ZZ domain on their N-terminus. This manual categorisation can be viewed as "clustering", as we group histones with similar predicted quaternary structures or with similar additional domains, although we don't explicitly refer to it as "clustering" in the manuscript. We performed sequence clustering primarily to generate an overview of what the sequence space looks like, not to categorise the histones. However, we noticed that a large cluster of histones which we initially defined as "undefined" was closely related in CLANS sequence clustering. We therefore decided to group these histones in a new category called "bacterial dimers", as they only appear in bacteria and don't have quaternary structural features which set them apart from other histones. This is the only category we defined based on sequence instead of structure. We added this information in the main text under the section "Identifying new prokaryotic histones" and in the Material and Methods under the section "Histone classification" to clarify this.

>More generally, it is difficult to tell what structure prediction is adding to the analysis. Which of the families from Figure 2a already have at least 1 crystal structure determined? For those cases, wouldn't homology modeling give the same level of detail of analysis (topology and putative binding residues & architecture)? While AlphaFold2 certainly gives more accurate predictions, it is not clear what that increased accuracy is adding to the overall conclusions. Are there specific examples where predicted structure was necessary for analysis? Or corrected something misclassified in the sequence-based clustering?

We agree with the reviewer that it was not explicitly clear in our original manuscript what structure prediction added to the analysis. Of all the histone categories visualised in Figure 2a, only 4 have crystal structures (Nucleosomal, Methanococcales, Bacterial dimer, and DUF1931), two of which have only appeared this year

[1, 2, 3]. For these 4 cases, homology modelling would work well if the sequence identity with the crystal structure is high enough, but many of our histones have extremely low sequence identity (<25%) with these crystal structures. However, even in the cases where homology modelling would work well, it still does not give as much information as AlphaFold2 does. For example, for histone MJ1647 from the Methanococcales category, there is a crystal structure available of the dimer (PDB: 8BDK). With homology modelling, we could predict the dimer structures of other Methanococcales histones. However, MJ1647 functions as a tetramer in solution. With homology modelling, we can not confidently predict this tetramer structure, whereas AlphaFold2 can produce a confident tetramer structure from the coevolutionary information in the MSA. AlphaFold2 tells us how these histones possibly multimerise, which will be closely linked to how these histones function and organise DNA. While all histones share the same monomeric histone fold, the main difference between all the prokaryotic histones is how they multimerise. Thus, the main contribution of AlphaFold2 is that it gives information about the multimerisation of these histones which is fundamental to how we categorise them. These predicted multimer structures in turn give more insight into the possible function of these histones than sequence alone. These insights are very valuable for future experimental work. For example, the tetrameric prediction of Methanococcales histone MJ1647 led us to hypothesise that it could bridge two DNA duplexes [1]. We could demonstrate experimentally that MJ1647 does indeed function as a DNA bridger [1]. Similarly, AlphaFold2 predicts that the bacterial dimer histone HBb/Bd0055 does not form larger oligomers than dimers. We hypothesised that it would function as a simple DNA bender; we recently experimentally confirmed this prediction in a preprint [4]. The structural predictions in our work are all testable and are expected to inspire and guide future experimental work by our lab as well as by others in the field.

>More information should be provided on the AlphaFold modeling. Of the 5800, how many had confident predictions? Were MSAs deep for all of them? Why were templates not used? (Did MSAs alone give very high-confidence predictions?) Were heteromeric predictions run at all? Are any of the histones believed to be heteromeric?

We agree that it is useful to provide more information on the AlphaFold2 modelling. We have now added histograms showing the distribution of MSA depth, pLDDT, and ipTM scores. The vast majority of MSAs are deep (99% have more than 100 sequences, 95% have more than 500). The same goes for the pLDDT scores, where 95% are above a pLDDT score of 70. Giving an overview of the confidence of interacting interfaces across the whole dataset is more difficult (the 2-dimensional predicted aligned error values). AlphaFold2 provides an interface predicted TM score (ipTM value). This value is dependent on the PAE values

across the whole chain. Since ipTM focuses on the whole chain instead of the interface residues, predictions with low ipTM values are not necessarily bad multimer predictions. We have therefore added the PAE plots for every prediction that we show in the manuscript, which provides confidence values for interacting residues, to show that all of the discussed multimer predictions have inter-chain interfaces with high confidence values. We have added text in the Supplementary section to explain how AlphaFold2 works, how to interpret the confidence values, and to discuss the MSA depth and confidence values from our dataset. We believe that this will aid the non-expert reader to better appreciate the quality and importance of our findings.

We decided to run AlphaFold2 without templates as templates in the majority of cases do not improve the quality of the prediction [5, 6]. We have now provided this reasoning for not using templates in the section "Identifying and predicting histones" in the Materials and methods section.

We only ran homomeric predictions. Some histones, such as the model histones HMfA and HMfB, are known to be able to form heteromers. However, HMfA and HMfB can also form homomeric complexes. Other histones, such as the recently described DNA-bridging histone MJ1647, can only form homomeric complexes. Thus, while some prokaryotic histones can form heteromers, all the prokaryotic histones studied to date can form and function as homomeric complexes. We decided not to run heteromeric predictions as this exponentially increases the number of predictions to be performed, to the point where it becomes unfeasible to do these predictions in a reasonable timeframe with the infrastructure available to us. We have added this reasoning for not running heteromeric predictions to the section "Identifying and predicting histones" in the Materials and methods section.

Reviewer 2

>Figure 2a, Do the authors have any comment or explanation of why TM and RdgC have disjunct distributions? Are the RdgC-like proteins in this figure?

This is indeed an interesting point. As we categorise the histones by structure and not by sequence (except for the bacterial dimer category; see above), categories can have disjunct groups in the CLANS sequence clustering. For RdgC histones, there are two disjunct groups in Figure 2A. The RdgC histones in both groups form the same tetramer structure (Figure 6C) and appear in the same genomic context (Supplementary Fig. 27 in the new manuscript version, S23 in the previous version). However, one group are RdgC histones from Archaea while the other group are from Bacteria. The sequence identity between the archaeal and bacterial groups is low (10%). Because the tetramer structure and the genomic context are identical, which implies they share some kind of function, we decided to categorise

them together as "RdgC histones". For the transmembrane histones, we observe very large sequence diversity, more than in any other category, hence the disjunct groups. We have no explanation for why the transmembrane histones differ so much in sequence.

>Page 4, bottom, "17 categories, 13 of which are new". This would be a good place to reference Table S1. In the Table, the authors could also mark the four previously known categories (maybe in bold font). For DUF1931 and Methanococcales they mentioned a known example from the class under discussion. They could do likewise for the bacterial dimer and nucleosomal categories.

This is a good suggestion. We added the changes to Table S1 and the reference on page 4.

>Page 5, top, the first mention of Figure S2 that I can find comes in the fold-to-fold section, after mentioning Figure S4. Did I miss something?

The reviewer is right. This has been corrected.

>Page 5, bottom, the numbering of amino acids in the text does not match the numbers in Figure 3, which is very confusing. In the legend to Figure 3 the correspondence between the numbers in the model of D4GZE0 and the Logos of the HMM model is listed, but is there a better way to do this? For example, after saying on page 5 that the conserved residues in the HMM are 48, 52, 54, 56, and 61, can the authors say these correspond to N45, R47, T49, and D54 in D4GZE0 and then use the latter numbers that match the figure?

This is indeed a very good suggestion. We have now implemented the suggested change in numbering throughout the manuscript.

>Also in Figure 3 Legend, please explain what pLDDT values are. The acronym is explained in the Methods, but people will read the figure legends first before they get to the Methods.

It is indeed important that figures are as easy to read as possible. We have now changed the Figure 3 legend. We have also added Supplementary Text which contains a general explanation of AlphaFold2, how to interpret the confidence metrics, and an overview of the confidence values and MSA depths of our whole dataset.

>In Figure 4 the amino acid numbers in the legend again do not match the figure and are off by 10. Please refer to the numbers that match the figure, and then you can mention the corresponding numbers in the logo in Figure S13 at the end of the legend.

This issue has now been corrected.

>In the legend to Figure S14 I suggest listing the organism after each protein (“DOLYZ1 from Haliangium ochraceum SMP-2 and E1WYC3 from Halobacteriovorax marinus SJ”). I was initially slightly confused by the way it is currently listed.

This is a good suggestion. We have now adapted the legend of Figure S14 accordingly.

>Page 8, end of second paragraph states that there is low expression of ZZ histone in both attack and growth phase, citing Figure S11; however, I only see ZZ marked in the attack phase in Figure S11b. Is there evidence for its abundance in the growth phase? Can it be marked in Figure S11a?

This is a valid point. The expression of the ZZ histone during the growth phase is so low that it does not show up in the top 2000 expressed genes. We have now changed the x-axis to show the top 3000 genes so that the ZZ histone now also appears in the figure. Proteomics data shows that the protein abundance of the ZZ histone is very low in the attack phases. In a strain which is host-independent, reflective of the growth phase, proteomics data does not show high protein abundance for the ZZ histone. We have added text elaborating on this under the section “Minor $\alpha 3$ histones” in the main text to refer to these proteomics data.

>Page 9, it is unclear why the conserved DNA-binding residues of bacteriophage histones are listed as R10 and R49 when the logo in Figure S17 implies K is much more common at positions 10 and 49.

The reviewer is correct. It should be K10 and K49. We have changed the text accordingly.

>The authors’ focus is on poorly known prokaryotic histones, but I wonder if in their discussion they want to compare to non-nucleosomal eukaryotic histones such as the TAFs with histone fold domains, at least some of which lack DNA-binding residues. A somewhat similar analysis of proteins with histone fold domains was recently published for plants (PMID: 33888032), with diverse functions for histone folds that might provide additional context for their findings.

This is a good suggestion. We primarily compare to prokaryotic nucleosomal histones as these are evolutionarily the closest group of well-studied histones to the novel histones we discuss. However, we agree that some of our novel histones, especially the ones without any conventional histone DNA-binding residues, could have evolved to gain functions not related to genome organisation. We have added some text to the discussion to highlight this point.

Reviewer 3

> My major concern is that the AlphaFold2 (AF2) method, which is deeply trained on the PDB structures ... may have structural bias in its predictions.

It is indeed true that AlphaFold2 is trained on structures from the Protein Database (PDB). This comes with the risk that AlphaFold2 is not a general structure predictor but instead acts as a look-up table of the PDB which would have strong structural biases. However, there are strong indications that AlphaFold2 has learned generalisable structural features of protein structure. The first indication is that AlphaFold2, the model which was trained only on individual protein chains in the PDB, can predict multimeric complexes with high accuracy [7]. This is a surprising result as it was never trained on any multimeric structure. The second indication is that AlphaFold2 is state-of-the-art at predicting intrinsically disordered regions [8, 9]. Again, this is something AlphaFold2 has not been trained on. Since the PDB consists only of well-structured proteins, we expect a bias to predict well-structured regions, but this bias is not observed. Third, AlphaFold2 is good at predicting and designing de novo proteins [10, 11]. Together, this shows that AlphaFold2 can generalize beyond its training dataset.

> Without experimental or further verification, the biological stories out of it are not convincing.

We agree with the reviewer that experimental verification of predictions is important. This study aims to identify novel histones and to construct a rudimentary view of how these histones possibly function to aid future experimental work on prokaryotic histones. From the predicted multimer structures, conserved residues, and published literature we construct hypotheses for how these histones could operate. The AlphaFold2 predictions have already been instrumental, for instance, in studying two new prokaryotic histones: MJ1647 from *Methanococcus jannaschii* and HBb/Bd0055 from *Bdellovibrio bacteriovorus*. For MJ1647 we hypothesised based on the predicted tetramer structure that it could bridge DNA. For HBb/Bd0055 we hypothesised based on the predicted dimer structure and the lack of confident predictions of higher oligomer structures from AlphaFold2 that it could function as a DNA bender. We were able to confirm both these hypotheses experimentally [1, 4]. Due to the sheer amount of predictions of novel histone functionality reported, in our view, the experimental verification lies outside the scope of this study. However, we firmly believe that our predictions provide excellent starting points for future experimental work by our lab and others in the field.

> AlphaFold2 ... typically uses template search for the best performance

The reviewer is correct: AlphaFold2 can indeed use templates as input for its predictions. However, there is little to no gain in performance for using templates [5, 6]. We therefore decided to not use templates for our predictions. Also, see our reply to reviewer 2.

>Also, since the query sequences that the authors used are from the result of homologous sequence search of histone-fold, it is not surprising that AF2 would predict the main histone-fold structure with shorter or longer local secondary structures at different places.

Indeed, we perform our initial search of histones based on sequence and thus one can reason that it is likely that we find histone folds. However, sequence identities between known histones and the prokaryotic histones that we discuss are often in or near the "twilight zone" (<25%). If the sequence identity is in the twilight zone, we cannot say with certainty if the two proteins are evolutionarily related and thus it is not obvious that the two proteins share the same protein fold [12, 13]. AlphaFold2 is therefore a powerful tool for us to check if these sequences likely form histone folds.

>Further, most of the predictions done in this work are among multiple histones as we know that histone monomer does not have stable structure. The AF2-complex is known to not have comparable performance vs. the AF2-monomer. Specific setups or processing are needed to enhance a good performance as has been shown in the recent CASP15 test (<https://doi.org/10.1016/j.sbi.2023.102594>).

As pointed out by the reviewer, AlphaFold2 Multimer indeed does perform worse than AlphaFold2, largely because the coevolutionary signal for multimerization is weaker than the signal for monomeric folding. In CASP15, some groups were able to improve the performance of AlphaFold2 Multimer by performing sampling, as explained in the article linked by reviewer 3 [6]. In sampling, you either try to create higher-quality MSAs or increase model diversity by enabling dropouts and generating massive amounts of models for each prediction. You then select the model which AlphaFold2 is most confident about as your final prediction. However, sampling is very computationally intensive. For example, AFSample, which performed better than AlphaFold2-Multimer in CASP15, required around 1000x more computational time per prediction than AlphaFold2-Multimer [14]. For the size of our database, sampling would be unfeasible to do in a reasonable timeframe with the infrastructure available to us. Importantly, however, in the vast majority of CASP15 multimer targets, AlphaFold2-Multimer generated predictions of equal quality to AFSample. Sampling is thus only practical for single difficult predictions where AlphaFold2-Multimer is uncertain about its prediction. However, our categorised histones have confident pLDDT and predicted aligned values (PAE)

values. To show these confidence values, we have added several plots as Supplementary information. We have added the distribution of multiple sequence alignments (MSA), pLDDT values, and ipTM values for every histone in our database. We have also included Supplementary Text to explain how AlphaFold2 works and how to interpret the pLDDT and PAE values. We have also added figures showing the PAEs of every multimer prediction we show in the manuscript to the Supplementary.

Relying on the confidence values from AlphaFold2, especially the PAE plots for the multimer predictions, is a good method for selecting high-quality predictions; PAE plots that show a confident multimer interface between chains (distance error $<10\text{\AA}$) can be considered high-quality multimer predictions. We use ColabFold, a community version of AlphaFold2 and AlphaFold2-Multimer, to predict the histone structures. In CASP15, ColabFold participated and predicted multimer structures of 41 targets (<https://wwwuser.gwdg.de/~mmirdit/casp15>). Out of the 41 targets, there are only two targets (H1144 and T1160) where the PAE plot shows a confident interface between the chains but the quaternary state (QS) score, which reflects the fraction of correctly modelled interface residues, was <0.3 , indicative of a low-quality multimer prediction (https://predictioncenter.org/casp15/results.cgi?tr_type=multimer). The multimer structures shown and discussed in the manuscript are all likely high-quality and correct predictions as the PAE plots contain inter-chain interfaces with low-distance errors.

>PIDDIT is a metric used in AF2 to describe the prediction confidence of the local regions. As another observation from the predicted structures, most of the interesting parts that support the authors points seem to have lower pLDDT than the main histone-fold frame.

The histones with additional domains, D0LYE7, A0A0F8XJF6, A0A2E7QIQ9, E3GZL0, D4GVY1, A0A358AGI2, A0A1F9E2M1, and A5UK87, have pLDDT values >70 for their additional domains and thus they can be considered as confident local structure predictions. In most of these histones, there are also residues with pLDDT values <50 . However, such low pLDDT values are generally viewed as predictions of disordered regions instead of as regions where AlphaFold2 fails to predict the relevant structures [8, 9].

References

- [1] S. Ofer, F. Blombach, A. M. Erkelens, D. Barker, Z. Soloviev, S. Schwab, K. Smollett, D. Matelska, T. Fouqueau, N. van der Vis, N. A. Kent, K. Thalassinou, R. T. Dame, and F. Werner. DNA-bridging by an archaeal his-

tone variant via a unique tetramerisation interface. *Communications Biology*, 6(1):1–16, Sept. 2023.

- [2] A. Hocher, S. P. Laursen, P. Radford, J. Tyson, C. Lambert, K. M. Stevens, A. Montoya, P. V. Shliaha, M. Picardeau, R. E. Sockett, K. Luger, and T. Warnecke. Histones with an unconventional DNA-binding mode in vitro are major chromatin constituents in the bacterium *Bdellovibrio bacteriovorus*. *Nature Microbiology*, pages 1–14, Oct. 2023.
- [3] K. Decanniere, A. M. Babu, K. Sandman, J. N. Reeve, and U. Heinemann. Crystal structures of recombinant histones HMfA and HMfB, from the hyperthermophilic archaeon *Methanothermus fervidus*. *Journal of Molecular Biology*, 303(1):35–47, 2000.
- [4] Y. Hu, S. Schwab, S. Deiss, P. Escudeiro, J. D. Joiner, M. D. Hartmann, A. N. Lupas, B. H. Alvarez, V. Alva, and R. T. Dame. Bacterial histone HBb from *Bdellovibrio bacteriovorus* compacts DNA by bending. *bioRxiv*, Sept. 2023.
- [5] J. Jumper, R. Evans, A. Pritzel, T. Green, M. Figurnov, O. Ronneberger, K. Tunyasuvunakool, R. Bates, A. Žídek, A. Potapenko, A. Bridgland, C. Meyer, S. A. Kohl, A. J. Ballard, A. Cowie, B. Romera-Paredes, S. Nikolov, R. Jain, J. Adler, T. Back, S. Petersen, D. Reiman, E. Clancy, M. Zielinski, M. Steinegger, M. Pacholska, T. Berghammer, S. Bodenstein, D. Silver, O. Vinyals, A. W. Senior, K. Kavukcuoglu, P. Kohli, and D. Hassabis. Highly accurate protein structure prediction with AlphaFold. *Nature*, 596(7873):583–589, 2021.
- [6] A. Elofsson. Progress at protein structure prediction, as seen in CASP15. *Current Opinion in Structural Biology*, 80:102594, June 2023.
- [7] P. Bryant, G. Pozzati, and A. Elofsson. Improved prediction of protein-protein interactions using AlphaFold2. *Nature Communications*, 13(1):1265, Mar. 2022.
- [8] K. Tunyasuvunakool, J. Adler, Z. Wu, T. Green, M. Zielinski, A. Žídek, A. Bridgland, A. Cowie, C. Meyer, A. Laydon, S. Velankar, G. J. Kleywegt, A. Bateman, R. Evans, A. Pritzel, M. Figurnov, O. Ronneberger, R. Bates, S. A. A. Kohl, A. Potapenko, A. J. Ballard, B. Romera-Paredes, S. Nikolov, R. Jain, E. Clancy, D. Reiman, S. Petersen, A. W. Senior, K. Kavukcuoglu, E. Birney, P. Kohli, J. Jumper, and D. Hassabis. Highly accurate protein structure prediction for the human proteome. *Nature*, 596(7873):590–596, Aug. 2021.

- [9] M. Akdel, D. E. V. Pires, E. P. Pardo, J. Jänes, A. O. Zalevsky, B. Mészáros, P. Bryant, L. L. Good, R. A. Laskowski, G. Pozzati, A. Shenoy, W. Zhu, P. Kundrotas, V. R. Serra, C. H. M. Rodrigues, A. S. Dunham, D. Burke, N. Borkakoti, S. Velankar, A. Frost, J. Basquin, K. Lindorff-Larsen, A. Bateman, A. V. Kajava, A. Valencia, S. Ovchinnikov, J. Durairaj, D. B. Ascher, J. M. Thornton, N. E. Davey, A. Stein, A. Elofsson, T. I. Croll, and P. Beltrao. A structural biology community assessment of AlphaFold2 applications. *Nature Structural & Molecular Biology*, 29(11):1056–1067, Nov. 2022.
- [10] J. Wang, S. Lisanza, D. Juergens, D. Tischer, J. L. Watson, K. M. Castro, R. Ragotte, A. Saragovi, L. F. Milles, M. Baek, I. Anishchenko, W. Yang, D. R. Hicks, M. Expòsit, T. Schlichthaerle, J.-H. Chun, J. Dauparas, N. Bennett, B. I. M. Wicky, A. Muenks, F. DiMaio, B. Correia, S. Ovchinnikov, and D. Baker. Scaffolding protein functional sites using deep learning. *Science*, 377(6604):387–394, July 2022.
- [11] B. I. M. Wicky, L. F. Milles, A. Courbet, R. J. Ragotte, J. Dauparas, E. Kinfu, S. Tipps, R. D. Kibler, M. Baek, F. DiMaio, X. Li, L. Carter, A. Kang, H. Nguyen, A. K. Bera, and D. Baker. Hallucinating symmetric protein assemblies. *Science*, 378(6615):56–61, Oct. 2022.
- [12] B. Rost. Twilight zone of protein sequence alignments. *Protein Engineering, Design and Selection*, 12(2):85–94, Feb. 1999.
- [13] S. Y. Chung and S. Subbiah. A structural explanation for the twilight zone of protein sequence homology. *Structure*, 4(10):1123–1127, Oct. 1996.
- [14] B. Wallner. AFsample: improving multimer prediction with AlphaFold using massive sampling. *Bioinformatics*, 39(9):btad573, Sept. 2023.

Reviewers' Comments:

Reviewer #1:

Remarks to the Author:

The authors answered my questions clearly, and addressed my concerns. Their explanation for modeling homomeric predictions only (and other experimental choices) seems reasonable.

However, the manuscript changes now make the 1st two paragraphs of the results section unnecessarily difficult to follow, as it is interspersing interpretation of results and methodology. Can this be reorganized in a way that better clarifies the experiments run?

Reviewer #2:

Remarks to the Author:

I am satisfied with the authors' response to my concerns. I appreciate the addition of the explanation of AlphaFold2 in the supplement and the additional information on the methodology.

I did notice a minor problem in the 2nd sentence of the Introduction: "fitting the relatively large chromosomal DNA into a cell that is orders of magnitude smaller." Smaller than what? The authors presumably mean that the cell diameter (or circumference) is orders of magnitude smaller than the linear length of the chromosomal DNA, but the chromosome and its condensed DNA clearly must be smaller than the cell. The authors can easily fix this.

Reviewer #3:

Remarks to the Author:

Again, this is a very interesting and comprehensive bioinformatics study and may be an inspiring and useful study for many biologists. In the submitted revision, I appreciate the authors' efforts in getting more supporting information for AlphaFold's capabilities in making predictions on 1). Multimer with model trainings only on monomer proteins; 2). IDR region predictions. First, it should be remembered that there are a number of IDR regions such as loops in the PDB structures, which I believe are not excluded particularly from the training set of AlphaFold. It is indeed feasible that AlphaFold has the capability to make new and correct structural predictions. However, as a pure application study in this work, it is not convincing enough to build main conclusions and stories including functionality hypothesis and clustering of protein families fundamentally based on AlphaFold predictions. Although the authors found interesting consistencies with previous experiments, as an independent work to be published on a prestigious journal like Nature Communications, I still think adding some new experimental results would be more convincing for the readers. I understand that the review process has been taking long, but it would really be helpful if the authors could design one or two experiments to confirm any of their computational findings. Moreover, considering the commonly accepted view in the field that most bacteria do not have histones, a title like "Novel histones and histone variant families in prokaryotes" without any experimental confirmations could be a challenge.

Two minor points are: 1) there are recent studies on histone folding across archaeal and eukaryotic species, revealing interesting conservations and evolutions of histones. The authors may consider citing and commenting on them accordingly.

2) since page 4, the authors have used the word "histones" to represent all the homologous protein sequences to be studied here, a quarter of which are from bacteria. Only in the discussions, it was noted those proteins might not be strictly called "histones". This reminder note could appear earlier to avoid confusion for experts in studying bacteria DNA binding proteins.

To all reviewers

All changes to the manuscript are highlighted in blue. As we have not yet finished depositing the plasmids made in this study, the AddGene numbers for these plasmids are not yet available in the supplementary information. In light of recent findings on bacterial histones BD055/HBb (updated preprint by Hu et al. (2024)), we have removed one of our proposed models for how face-to-face (previously called fold-to-fold) histones bind DNA (Figure 4) as this new data disfavors the now removed model.

Reviewer 1

>However, the manuscript changes now make the 1st two paragraphs of the results section unnecessarily difficult to follow, as it is interspersing interpretation of results and methodology. Can this be reorganized in a way that better clarifies the experiments run?

This is a good point. We have reduced the amount of methodology in the 1st two paragraphs.

Reviewer 2

>I did notice a minor problem in the 2nd sentence of the Introduction: "fitting the relatively large chromosomal DNA into a cell that is orders of magnitude smaller." Smaller than what? The authors presumably mean that the cell diameter (or circumference) is orders of magnitude smaller than the linear length of the chromosomal DNA, but the chromosome and its condensed DNA clearly must be smaller than the cell. The authors can easily fix this.

Thank you for the comment. We have clarified this statement in the introduction.

Reviewer 3

>It is indeed feasible that Alphafold has the capability to make new and correct structural predictions. However, as a pure application study in this work, it is not convincing enough to build main conclusions and stories including functionality hypothesis and clustering of protein families fundamentally based on Alphafold predictions. Although the authors found interesting consistencies with previous experiments, as an independent work to be published on a prestigious journal like Nature Communications, I still think adding some new experimental results would

be more convincing for the readers. I understand that the review process has been taking long, but it would really be helpful if the authors could design one or two experiments to confirm any of their computational findings. Moreover, considering the commonly accepted view in the field that most bacteria do not have histones, a title like “Novel histones and histone variant families in prokaryotes” without any experimental confirmations could be a challenge.

As requested by the reviewer, we have performed two experiments to confirm our computational results. We have purified the face-to-face (previously called fold-to-fold) histone from *Thermococcus kodakarensis*, referred to as HTkC, and performed X-ray crystallography. We have solved the crystal structure of HTkC to a resolution of 1.84 Å. The crystal structure of HTkC is identical to the tetrameric AlphaFold prediction with a C α RMSD between the two structures of 0.652 Å, highlighting AlphaFold’s accuracy in predicting structures of novel histones. We hypothesized in the manuscript that coiled-coil histones bridge DNA based on their predicted tetramer structure. To test this, we have purified the coiled-coil histone from *Methanothermus fervidus*, referred to as HMfC, and performed a DNA-bridging assay. In the assay, we observe an increase in DNA-bridging activity as a function of HMfC’s concentration, showing that HMfC indeed bridges DNA, highlighting how AlphaFold can be used to gain accurate insights into the DNA-binding properties of novel histones. These results have been added to the manuscript in the results section, Figure 4, and Supplementary Figure 25.

> There are recent studies on histone folding across archaeal and eukaryotic species, revealing interesting conservations and evolutions of histones. The authors may consider citing and commenting on them accordingly.

This is a good suggestion. We have added the recent findings from Zhao et al. (2024) on histone folding and the effects of the eukaryotic tails to the introduction.

> since page 4, the authors have used the word “histones” to represent all the homologous protein sequences to be studied here, a quarter of which are from bacteria. Only in the discussions, it was noted those proteins might not be strictly called “histones”. This reminder note could appear earlier to avoid confusion for experts in studying bacteria DNA binding proteins.

This is a good point. We have clarified our definition of a ”histone” in the ”Identifying new prokaryotic histones” section.

Reviewers' Comments:

Reviewer #2:

Remarks to the Author:

The authors appear to me to have satisfied all of the comments of the reviewers.

Reviewer #3:

Remarks to the Author:

The revised manuscript added two experiments to test their computational hypotheses. They crystalized one protein structure. Although AF2-predicted structure was used as the search model to determine the phases of X-ray data, given the known conservation of histone-fold family, it is reasonable. The crystalized structure (Fig. 3d) shows clear parallel beta-sheets of loops, which is one of the structure signatures observed in many histone and nucleosome structures. The authors may consider mentioning this experiment in the abstract.

The other experiment conducted DNA pull-down assay with HMfC protein, which AF2 showed to have a tetramer structure thus may bridge two DNA sites on its two ends. The pulled-down DNA increases almost linearly with the HMfC concentration (Fig S25), indicating the binding between HMfC and DNA. However, would the dimerized HMfc also give similar result? Could the authors discuss it?

Overall, the revision addressed my major concerns with only a minor question and suggestion as described above.

Reviewer 3

> *The revised manuscript added two experiments to test their computational hypotheses. They crystalized one protein structure. Although AF2-predicted structure was used as the search model to determine the phases of X-ray data, given the known conservation of histone-fold family, it is reasonable. The crystalized structure (Fig. 3d) shows clear parallel beta-sheets of loops, which is one of the structure signatures observed in many histone and nucleosome structures. The authors may consider mentioning this experiment in the abstract.*

We thank the reviewer for their comment. We do not explicitly mention the crystallization in the abstract as the focus of the manuscript is more on computational methods. The crystallization of HTkC and the bridging assay of HMfC were performed to support the computational data.

> *The other experiment conducted DNA pull-down assay with HMfC protein, which AF2 showed to have a tetramer structure thus may bridge two DNA sites on its two ends. The pulled-down DNA increases almost linearly with the HMfC concentration (Fig S25), indicating the binding between HMfC and DNA. However, would the dimerized HMfC also give similar result? Could the authors discuss it?*

We thank the reviewer for their comment. We expect that a mutant of HMfC that can only dimerize will result in a loss of bridging. We have done similar experiments with bridging histone MJ1647 whereby loss of tetramerization resulted in the loss of bridging [1] Creating and testing this HMfC mutant lies outside of the scope of this manuscript. However, we are working on a follow-up manuscript that will further characterize HMfC.

References

- [1] Sapir Ofer, Fabian Blombach, Amanda M. Erkelens, Declan Barker, Zoja Soloviev, Samuel Schwab, Katherine Smollett, Dorota Matelska, Thomas Fouqueau, Nico van der Vis, Nicholas A. Kent, Konstantinos Thalassinos, Remus T. Dame, and Finn Werner. DNA-bridging by an archaeal histone variant via a unique tetramerisation interface. *Communications Biology*, 6:1–16, September 2023.